# Dynamic Gradient Alignment for Online Data Mixing

## Abstract

The composition of training data mixtures is critical for effectively training large language models (LLMs), as it directly impacts their performance on downstream tasks. Our goal is to identify an optimal data mixture to specialize an LLM for a specific task with access to only a few examples. Traditional approaches to this problem include ad-hoc reweighting methods, importance sampling, and gradient alignment techniques. This paper focuses on gradient alignment and introduces Dynamic Gradient Alignment (DGA), a scalable online gradient alignment algorithm. DGA dynamically estimates the pre-training data mixture on which the models' gradients align as well as possible with those of the model on the specific task. DGA is the first gradient alignment approach that incurs minimal overhead compared to standard pre-training and outputs a competitive model, eliminating the need for retraining the model. Experimentally, we demonstrate significant improvements over importance sampling in two key scenarios: (i) when the pre-training set is small and importance sampling overfits due to limited data; and (ii) when there is insufficient specialized data, trapping importance sampling on narrow pockets of data. Our findings underscore the effectiveness of gradient alignment methods in optimizing training data mixtures, particularly in data-constrained environments, and offer a practical solution for enhancing LLM performance on specific tasks with limited data availability.

## 1 Introduction

Large Language Models (LLMs) are typically pre-trained on extensive, generic corpora sourced from a variety of data domains (Brown et al., 2020b; Touvron et al., 2023; Zhang et al., 2022), with the composition of these corpora often depending on domain availability or heuristics (Gao et al., 2020; Together AI Team, 2023). While the diversity of natural texts allows the model to learn from various knowledge sources, not all data domains are equally beneficial according to the targeted tasks. The uncurated nature of web-crawled contents could lead to sub-optimal outcomes due to the variations in data quality (Longpre et al., 2023). Plus, some domains may contain misinformation and biases, as one potential source of hallucinations in language generation (Lin et al., 2022; Huang et al., 2023).

To better generalize to the downstream target tasks, it is critical to identify the most beneficial subset from large, generic pretraining corpora. While sample-level selection can be costly, *domain reweighting* offers an efficient group-level approach. Domain reweighting methods assume that samples from the same domain share similar features and search for optimal sampling weights across *domains* (Xie et al., 2023a; Fan et al., 2024; Liu et al., 2024; Kang et al., 2024; Grangier et al., 2024). The domains that most positively impact the target tasks should be assigned higher weights.

In this work, on top of a large, generic pretraining corpus, we assume we have access to a few examples representative of the downstream task on which we want the model to generalize, a so-called *specialized set*. For this setup, Grangier et al. (2024) recently proposed a simple and scalable *importance sampling* based method to domain reweighting, where the weight of a domain is given by the frequency of samples in the specialized set closest to the domain, where distance is measured with SentenceBert (Reimers & Gurevych, 2019) embeddings. This method determines the domain weights before any training and is model-agnostic.

Likewise, prior gradient-alignment methods determine a ***static*** domain weights for large-scale LM training, often relying on a small-scale proxy model (Xie et al., 2023a; Fan et al., 2024) or fitting a scaling law (Liu et al., 2024; Kang et al., 2024). While these methods show improvements over training on the natural distribution of a generic corpus, they do not dynamically update domain

weights during training to adapt to the current model state. In practical training scenarios, a large model may quickly overfit on certain domains with high weights. In such cases, an online weighting method can respond by shifting emphasis to other domains.

We propose Dynamic Gradient-Alignment (DGA), an ***online*** domain reweighting method that estimates step-wise optimal domain weights during model training. Inspired by DoGE (Fan et al., 2024), at each reweighting step, DGA upweights the data domain whose gradient aligns more with the model's gradient on the specific set. From the optimization perspective, training the model on the most-aligned data domain yields the greatest reduction in the targeted loss. By incorporating an exponential-moving-average (EMA) term in online domain weights updates, DGA effectively mitigates overfitting and prioritizes the domains that currently benefit the target task the most. Since the domain weights and model parameters are updated concurrently, inaccurate domain weights can potentially drive the model into suboptimal states, which further leads to snow-balled errors. In such cases, the EMA term serves as a correction factor, guiding the model back to a more stable state. As an additional contribution, we scale the domain reweighting methods into extremely fine-grained domains (e.g. $262k$ domains) by introducing a novel *distribution reweighting mechanism*. Rather than directly reweighting $262k$ data domains, distribution reweighting reparameterizes the high-dimensional domain weights as a convex combination of weight vectors derived from a set of distributions estimated from embedding-based importance sampling (Grangier et al., 2024). With the number of distributions less than the number of training data domains, it allows DGA to scale to thousands of domains and make the most of the fine-grained group-level features.

Our experiments demonstrate the effectiveness of DGA compared to standard pre-training and importance sampling baselines in two challenging cases: (1) the resource of training tokens in each domain is limited instead of infinite (§ 3.1), and (2) the domain granularity is extremely large, which introduces intractable computation overheads on the domain reweighting problem (§ 3.2).

## 2 DATA MIXING WITH SPECIALIZED TARGET

### 2.1 GENERIC DATASET AND SPECIFIC TASKS

We consider a generic training corpus $D_{\text{gen}} = \{D_1, \ldots, D_k\}$, which is partitioned into $k$ distinct data domains. We can sample from each of the $k$ domains to train a model. Consequently, we can sample from a *mixture* of these domains and draw a batch of data following the law $\boldsymbol{x} \sim \text{mix}(\boldsymbol{\alpha}) \triangleq \sum_{i=1}^{k} \alpha_i D_i$, where $\boldsymbol{\alpha} \in \mathbb{R}^k$ is the mixture *weights*, belonging to the *simplex* $\boldsymbol{\alpha} \in \Delta^k \triangleq \{\boldsymbol{\alpha} \in \mathbb{R}^k | \sum_{i=1}^{k} \alpha_i = 1 \text{ and } \alpha_i \geq 0 \text{ for all } i\}$. Here, getting one sample from $\text{mix}(\boldsymbol{\alpha})$ means first getting a random index $i \in \{k\}$ from the categorical distribution corresponding to the vector of probabilities $\boldsymbol{\alpha}$, and then outputting a random sample from the domain $D_i$. Sampling from this law is computationally efficient if we can efficiently sample from each domain. Next, we consider a model, parameterized by $\boldsymbol{\theta} \in \mathbb{R}^p$, and a loss function $\ell(\boldsymbol{\theta}, \boldsymbol{x})$ defined for $\boldsymbol{x} \in D_{\text{gen}}$. To simplify notation, given a set of samples $S$ (which can be either a full dataset $D_i$, or a mini-batch), we denote the average over $S$ of the loss $\ell(\boldsymbol{\theta}, S) \triangleq \frac{1}{\#S} \sum_{\boldsymbol{x} \in S} \ell(\boldsymbol{\theta}, \boldsymbol{x})$. Since we focus on LLMs, $\ell$ is typically the next-token-prediction loss. For a given mixture weight $\boldsymbol{\alpha}$, we can update $\boldsymbol{\theta}$ by doing optimization steps on the *generic loss*

$$L_{\text{gen}}(\boldsymbol{\theta}, \boldsymbol{\alpha}) \triangleq \mathbb{E}_{\boldsymbol{x} \sim \text{mix}(\boldsymbol{\alpha})}[\ell(\boldsymbol{\theta}, \boldsymbol{x})] = \sum_{i=1}^{k} \alpha_i L_i(\boldsymbol{\theta}) \text{ with } L_i(\boldsymbol{\theta}) \triangleq \ell(\boldsymbol{\theta}, D_i) \quad (1)$$

In this paper, our goal is to use this data-mixture to train a model that performs well a *specific* task. We assume to have access to samples from this task, split into train and test sets. We call the train set the *specific dataset* $D_{\text{spe}}$ that we use to train models. The performance on the specific set is measured with the *specific loss*

$$L_{\text{spe}}(\boldsymbol{\theta}) \triangleq \ell(\boldsymbol{\theta}, D_{\text{spe}}). \quad (2)$$

We assume that the specific set $D_{\text{spe}}$ is small; hence, optimizing $L_{\text{spe}}$ directly leads to overfitting: the loss on the test data would be much higher than $L_{\text{spe}}$. Instead, to get to a low specific loss, we aim to find the optimal data mixing across $k$ data domains $\boldsymbol{\alpha}$ at each training step to get a good model while training on the reweighted generic distribution $\text{mix}(\boldsymbol{\alpha})$.

The target specialization task can be flexible according to the application domains, ranging from reasoning, instruction following, etc., corresponding to various objective loss functions, including the

next-token prediction loss and preference-based losses when applied on pair-wise datasets. In this paper, we focus on next-token prediction on another dataset. Next, we introduce a general bilevel formulation of the data mixing problem.

## 2.2 BILEVEL FORMULATION

Since the lack of data forbids optimizing directly $L_{\mathrm{spe}}$, we look for the mixture $\boldsymbol{\alpha}$ such that optimizing the generic loss $L_{\mathrm{gen}}(\boldsymbol{\theta}, \boldsymbol{\alpha})$ yields the smallest specific loss (Grangier et al., 2023). This is formalized by the following *bilevel optimization* problem (Bracken & McGill, 1973; Dagréou et al., 2022):

$$\boldsymbol{\alpha}^{\star} \in \arg\min_{\boldsymbol{\alpha} \in \Delta^k} L_{\mathrm{spe}}(\boldsymbol{\theta}^{\star}(\boldsymbol{\alpha})), \text{ such that } \boldsymbol{\theta}^{\star}(\boldsymbol{\alpha}) \in \arg\min_{\boldsymbol{\theta}} L_{\mathrm{gen}}(\boldsymbol{\theta}, \boldsymbol{\alpha}) \tag{3}$$

This bilevel formulation is intuitive: for a given weight $\boldsymbol{\alpha}$, the parameters obtained by minimizing the generic loss $L_{\mathrm{gen}}$ are $\boldsymbol{\theta}^*(\boldsymbol{\alpha})$, and we want those weights to yield a small specific loss. Notably, if the specific loss is a mixture of generic data with an unknown weight $\tilde{\boldsymbol{\alpha}}$, the bilevel formulation is guaranteed to recover it. In other words:

**Theorem 1.** *Assume that there exists $\tilde{\boldsymbol{\alpha}}$ such that $D_{\mathrm{spe}} = \mathrm{mix}(\tilde{\boldsymbol{\alpha}})$ . Then, $\tilde{\boldsymbol{\alpha}}$ is a solution to the bilevel problem in Equation 3.*

*Proof.* We let $\tilde{\boldsymbol{\theta}}$ the minimizer of $L_{\mathrm{spe}}$. Then, for all $\boldsymbol{\alpha}$, we have by definition that $L_{\mathrm{spe}}(\boldsymbol{\theta}^*(\boldsymbol{\alpha})) \geq L_{\mathrm{spe}}(\tilde{\boldsymbol{\theta}})$. Furthermore, since $D_{\mathrm{spe}} = \mathrm{mix}(\tilde{\boldsymbol{\alpha}})$, we have that $L_{\mathrm{gen}}(\boldsymbol{\theta}, \tilde{\boldsymbol{\alpha}}) = L_{\mathrm{spe}}(\boldsymbol{\theta})$ for all $\boldsymbol{\theta}$, hence minimizing this yields $\boldsymbol{\theta}^*(\boldsymbol{\alpha}) = \tilde{\boldsymbol{\theta}}$. Putting these results together, we have proven that for all $\boldsymbol{\alpha}$, it holds $L_{\mathrm{spe}}(\boldsymbol{\theta}^*(\boldsymbol{\alpha})) \geq L_{\mathrm{spe}}(\boldsymbol{\theta}^*(\tilde{\boldsymbol{\alpha}}))$, so that $\tilde{\boldsymbol{\alpha}}$ is a solution to Equation 3. □

We consider two types of methods to solve Equation 3. Static methods construct a single mixture weight vector $\boldsymbol{\alpha}$ and then minimize $L_{\mathrm{gen}}(\boldsymbol{\theta}, \boldsymbol{\alpha})$; we describe in the next section how to obtain this vector $\boldsymbol{\alpha}$. Online methods modify the weights dynamically during model training. They produce a sequence of weights $\boldsymbol{\alpha}^{(t)}$ where $t$ is the optimization iterate. In that case, at each training step, the parameters $\boldsymbol{\theta}^{(t)}$ are updated by doing an optimization step — with gradient descent or Adam — on the function $L_{\mathrm{gen}}(\boldsymbol{\theta}, \boldsymbol{\alpha}^{(t)})$. We now discuss methods to obtain a weight vector $\boldsymbol{\alpha}$ or a sequence $\boldsymbol{\alpha}^{(t)}$.

## 2.3 A STRONG BASELINE: IMPORTANCE SAMPLING

A sensible strategy is to train the model on a data mixture that most resembles the composition of the targeted specialization data distribution. This is the philosophy behind importance sampling (Kloek & Van Dijk, 1978). We estimate the importance sampling weights $\boldsymbol{\alpha}^{\mathrm{IS}}$ using the method of Grangier et al. (2024). The core idea is to embed each generic domain using SentenceBert (Reimers & Gurevych, 2019), and then compute the centroid of each domain $\boldsymbol{b}_i = \frac{1}{\#D_i} \sum_{\boldsymbol{x} \in D_i} \mathrm{Bert}(\boldsymbol{x})$. This defines a simple and cheap to compute selection function $c(\boldsymbol{x}) \in \{1 \dots k\}$, assigning $\boldsymbol{x}$ to its closest centroid, i.e., $c(\boldsymbol{x}) = \arg\min_i \|\mathrm{Bert}(\boldsymbol{x}) - \boldsymbol{b}_i\|$ for $\boldsymbol{x} \in D_{\mathrm{gen}} \cup D_{\mathrm{spec}}$. We use it to predict the closest generic data domain for each data instance from the specific set. The importance sampling weights are obtained as the ratio of examples falling in each bin: $\boldsymbol{\alpha}_i^{\mathrm{IS}} \triangleq \frac{\#\{\boldsymbol{x} \in D_{\mathrm{spe}} | c(\boldsymbol{x})=i\}}{\#D_{\mathrm{spe}}}$. One of the main advantages of this method is its simplicity: the computation of the weights $\boldsymbol{\alpha}^{\mathrm{IS}}$ is decoupled from model optimization and can be performed before training. It is expected to work well when the specialization set can be well approximated by the reweighted generic set, i.e., when $L_{\mathrm{spe}}(\boldsymbol{\theta}) \simeq L_{\mathrm{gen}}(\boldsymbol{\theta}, \boldsymbol{\alpha}^{\mathrm{IS}})$. When this is not the case, it might not lead to a good specific loss. Another potential issue with this method arises when it assigns a large weight to a generic domain $D_i$ with little available data. In this case, training a model on $\mathrm{mix}(\boldsymbol{\alpha}^{\mathrm{IS}})$ will overfit on that domain $D_i$, and it would have been better to reduce the weight of that domain to mitigate overfitting. A last issue arises when the number of specific examples, $\#D_{\mathrm{spe}}$, is significantly smaller than the number of domains $k$. In this situation, the importance weights become sparse, as they can have at most $\#D_{\mathrm{spe}}$ non-zero coefficients. This sparsity could be problematic, as some domains with zero weights might still be close to $D_{\mathrm{spe}}$. We illustrate these shortcomings in our experiments and explain how gradient alignment methods — which we introduce next — overcome them.

## 2.4 DGA: DYNAMIC GRADIENT ALIGNMENT

**Algorithm.** We introduce the DGA: Dynamic Gradient Alignment method for data reweighting to approximately solve the bilevel problem in Equation 3. This algorithm builds upon DoGE (Fan et al., 2024) and we give a precise account of their differences later. DGA keeps track of the model's

parameters $\boldsymbol{\theta}^t$ and dynamic weights $\boldsymbol{\alpha}^t$. Once every $T_r$ steps, we compute the gradient alignments $\boldsymbol{a}^t$, by doing

$$\boldsymbol{a}_i^t = \langle \nabla\ell(\boldsymbol{\theta}^t, \boldsymbol{x}_i), \nabla\ell(\boldsymbol{\theta}^t, \boldsymbol{z}) \rangle \text{ where } \boldsymbol{x}_i \sim D_i \text{ and } \boldsymbol{z} \sim D_{\text{spe}}. \tag{4}$$

and update the weights by mirror descent on the simplex (Beck & Teboulle, 2003) with step $\eta > 0$:

$$\boldsymbol{\alpha}^{t+1} = \frac{\hat{\boldsymbol{\alpha}}}{\sum_{i=1}^k \hat{\boldsymbol{\alpha}}_i} \text{ where } \hat{\boldsymbol{\alpha}} = \boldsymbol{\alpha}^t \odot \exp(\eta \boldsymbol{a}^t) \tag{5}$$

We optionally store an EMA version of the weights $\boldsymbol{\alpha}^t$ parameterized by $\beta \in [0, 1]$ to stabilize the training dynamics of the model's parameters, and define $\boldsymbol{\alpha}_{\text{EMA}}^{t+1} = (1 - \beta)\boldsymbol{\alpha}_{\text{EMA}}^t + \beta\boldsymbol{\alpha}^{t+1}$. Finally, at each step, we update the model's parameters $\boldsymbol{\theta}^t$ by doing an optimization step on $L_{\text{gen}}(\boldsymbol{\theta}, \boldsymbol{\alpha}_{\text{EMA}}^t)$. The full algorithm pseudo-code is given in Algorithm 1.

**Rationale.** This algorithm can be seen as a heuristic to solve the bilevel problem in Equation 3. Indeed, each update on $\boldsymbol{\theta}$ optimizes the inner loss. The update rule on $\boldsymbol{\alpha}$ can be seen as a mirror-descent step on $L_{\text{spe}}(\boldsymbol{\theta}^*(\boldsymbol{\alpha}))$ with several approximations. The first approximation consists of approximating the solution of the inner problem with one gradient descent step with step-size $\rho$: $\boldsymbol{\theta}^*(\boldsymbol{\alpha}) \simeq \boldsymbol{\theta}^t - \rho\sum_{i=1}^k \boldsymbol{\alpha}_i \nabla L_i(\boldsymbol{\theta}^t)$. We then approximate the specific loss at $\boldsymbol{\theta}^*$ by the post-update specific loss function: $L_{\text{spe}}(\boldsymbol{\theta}^*(\boldsymbol{\alpha})) \simeq f(\boldsymbol{\alpha}, \rho) \triangleq L_{\text{spe}}(\boldsymbol{\theta}^t - \rho\sum_{i=1}^k \boldsymbol{\alpha}_i \nabla L_i(\boldsymbol{\theta}^t))$, that is, the drop on the specific loss after an update. When the step size $\rho$ is small, a Taylor expansion gives

$$f(\boldsymbol{\alpha}, \rho) = L_{\text{spe}}(\boldsymbol{\theta}^t) - \rho\sum_{i=1}^k \boldsymbol{\alpha}_i \langle \nabla L_i(\boldsymbol{\theta}^t), \nabla L_{\text{spe}}(\boldsymbol{\theta}^t) \rangle + o(\rho) \tag{6}$$

Similarly, we get that the gradient of $f$ is the gradient alignment:

$$\frac{\partial f}{\partial \boldsymbol{\alpha}_i}(\boldsymbol{\alpha}, \rho) = -\rho\langle \nabla L_i(\boldsymbol{\theta}^t), \nabla L_{\text{spe}}(\boldsymbol{\theta}^t) \rangle + o(\rho) \tag{7}$$

We want to use this gradient of $f$ to implement a mirror-descent method. Unfortunately, the gradients involved in the alignment are full-batch, so we approximate them with stochastic gradients obtained from mini-batches, yielding the alignments $\boldsymbol{a}^t$ from Equation 4. Overall, we get the approximation $\nabla_{\boldsymbol{\alpha}} L_{\text{spe}}(\boldsymbol{\theta}^*(\boldsymbol{\alpha})) \simeq -\rho\boldsymbol{a}^t$; and the update rule in Equation 5 is a mirror descent step with this approximated gradient and step $\eta/\rho$.

We have explained the link between our algorithm and the bilevel problem in Equation 3. Proofs showing convergence of our method require assumptions violated in practice, e.g. most theoretical work assumes that the function $\boldsymbol{\theta} \to L_{\text{gen}}(\boldsymbol{\theta}, \boldsymbol{\alpha})$ is convex (Ghadimi & Wang, 2018; Arbel & Mairal, 2021; Dagréou et al., 2022). Nevertheless, successful applications of related bilevel algorithms to non-convex neural networks have been reported recently (Fan et al., 2024; Grangier et al., 2023).

**Computational cost and memory overhead.** The computation cost of DGA is compared to the cost of a regular pre-training run. For a base run iteration, the main cost is $t_g$, the cost of computing a gradient with a mini-batch $B$. For DGA, we need to add the cost of updating the domain weights $\boldsymbol{\alpha}$, which only happens every $T_r$ iterations. This update requires computing the $k + 1$ gradients (one per domain, one for $L_{\text{spe}}$). Hence the average cost of one iteration of DGA is $(1 + (k + 1)T_r^{-1})t_g$. Therefore, DGA's compute overhead is small when $T_r$ is large compared to the number of domains $k$.

During training, the memory is used by the optimizer state, the model gradients and its activations. We assume the same precision for storing all vectors. The optimizer state (the model parameters and the two EMA terms for Adam) and the gradients have a storage cost of $4m_g$, where $m_g$ denotes the cost of storing the model parameters. The cost of storing the activations during backpropagation is $m_b$. Regular pretraining with Adam therefore costs $4m_g + m_b$. DGA computes the required gradients sequentially and does not require more memory to store activations. It simultaneously stores two gradients instead of one (one domain gradient and one specific gradient). DGA, therefore, costs $5m_g + m_b$: DGA memory overhead ranges from 0 (when $m_b \gg m_g$) to $25\%$ (when $m_g \gg m_b$).

**Comparison with DOGE.** While our method is heavily inspired by DoGE (Fan et al., 2024), there are several key differences. First, DGA samples from the mixture: the weights $\boldsymbol{\theta}^t$ are updated using samples drawn from the mixture $\text{mix}(\boldsymbol{\alpha}^t)$, with the gradient $\nabla\ell(\boldsymbol{\theta}^t, \boldsymbol{x})$ where $\boldsymbol{x} \sim \text{mix}(\boldsymbol{\alpha}^t)$; this is the same gradient that one would use during pre-training with weight $\boldsymbol{\alpha}^t$. In contrast, DoGE's weights are updated using a reweighted gradient $\sum_{i=1}^k \boldsymbol{\alpha}_i^t \nabla\ell(\boldsymbol{\theta}^t, \boldsymbol{x}_i)$, where each $\boldsymbol{x}_i$ are drawn from

---

**Algorithm 1** Dynamic Gradient Alignment method

---

1: **Input:** Generic domains $D_1, \ldots, D_k$, specific set $D_{\text{spe}}$, inner optimizer state $\boldsymbol{\omega}^0$, optimizer function `Optimizer` such as Adam or SGD, initial weights $\boldsymbol{\alpha}^0$, outer learning rate $\eta$, EMA parameter $\beta$, weight update frequency $T_r$
2: **Initialize EMA weights:** $\boldsymbol{\alpha}_{\text{EMA}}^0 = \boldsymbol{\alpha}^0$
3: **for** $t = 0 \ldots T$ **do**
4:      Sample a batch from EMA generic mixture: $\boldsymbol{x} \sim \text{mix}(\boldsymbol{\alpha}_{\text{EMA}}^t)$
5:      Update the parameters $\boldsymbol{\theta}^{t+1}, \boldsymbol{\omega}^{t+1} \leftarrow \texttt{Optimizer}(\boldsymbol{\theta}^t, \boldsymbol{\omega}^t, \nabla_{\boldsymbol{\theta}}\ell(\boldsymbol{\theta}^t, \boldsymbol{x}))$
6:      **if** $t\%T_r = 0$ **then**
7:          Sample a batch from each domain: $\boldsymbol{x}_i \sim D_i$ for $i = 1 \ldots k$ and $\boldsymbol{y} \sim D_{\text{spe}}$
8:          Compute gradient alignements $\boldsymbol{a}_i^t \leftarrow \langle \nabla\ell(\boldsymbol{\theta}^{t+1}, \boldsymbol{x}_i), \nabla\ell'(\boldsymbol{\theta}^{t+1}, \boldsymbol{y}) \rangle$
9:          Update instantaneous weights: $\boldsymbol{\alpha}^{t+1} \leftarrow \frac{\hat{\boldsymbol{\alpha}}}{\sum_{i=1}^k \hat{\boldsymbol{\alpha}}_i}$ with $\hat{\boldsymbol{\alpha}} = \boldsymbol{\alpha}^t \odot \exp(-\eta\boldsymbol{a}^t)$
10:         Update EMA weights: $\boldsymbol{\alpha}_{\text{EMA}}^{t+1} \leftarrow \beta\boldsymbol{\alpha}_{\text{EMA}}^t + (1-\beta)\boldsymbol{\alpha}^{t+1}$
11:      **else**
12:          Do nothing: $\boldsymbol{\alpha}_{\text{EMA}}^{t+1} \leftarrow \boldsymbol{\alpha}_{\text{EMA}}^t$, and $\boldsymbol{\alpha}^{t+1} \leftarrow \boldsymbol{\alpha}^t$
13:      **end if**
14: **end for**
15: **Return** Optimized parameters $\boldsymbol{\theta}^{(T)}$ and weights trajectory $\boldsymbol{\alpha}^t, t = 0 \ldots T$

---

the domain $D_i$. For a fixed number of samples available at each draw, DGA's gradient estimate has a lower variance (Seiffert et al., 2008). As explained above, DGA has a small overhead compared to regular pre-training, while DoGE updates the weights at each iteration. These two key differences mean that DGA is much closer to regular pre-training than DoGE. For instance, DGA never requires retraining a model from scratch using the mixture weights estimated from a previous run, while this is the costly strategy used for DoGE. Finally, the EMA strategy described above is novel.

**Towards a convergence theory for DGA.** It is hard to prove the convergence of algorithms for data reweighting with neural networks because of the non-convexity of the loss functions and the unknown link between generic and specialist datasets. We prove the convergence of DGA in a simplified setting where the generic losses are deterministic quadratic functions, and the specialist dataset is exactly a mixture of generic datasets with unknown proportions $\tilde{\boldsymbol{\alpha}}$. We have:

**Theorem 2.** *Let* $\boldsymbol{\mu}_1, \ldots \boldsymbol{\mu}_k \in \mathbb{R}^d$ *some target vectors, and define the losses on the* $i^{th}$ *generic domain as* $L_i(\boldsymbol{\theta}) = \frac{1}{2}\|\boldsymbol{\theta} - \boldsymbol{\mu}_i\|^2$. *Let* $\tilde{\boldsymbol{\alpha}} \in \Delta_k$ *a target mixture vector, and assume that the specific loss is* $L_{\text{spe}}(\boldsymbol{\theta}) = \sum_{i=1}^k \tilde{\alpha}_i L_i(\boldsymbol{\theta})$. *Let* $M = [\boldsymbol{\mu}_1, \ldots, \boldsymbol{\mu}_k] \in \mathbb{R}^{d \times k}$, *assume that* $\lambda_{\min}(M^T M) > 0$. *Then, running DGA for* $T$ *iterations with gradient descent as the optimizer function with step size 1, and outer learning rate* $\eta = O(1/\sqrt{T})$ *yields iterates* $\boldsymbol{\alpha}^t$ *such that* $\min_{t=1\ldots T} \|\boldsymbol{\alpha}^t - \tilde{\boldsymbol{\alpha}}\|^2 = O(1/\sqrt{T})$.

This theorem demonstrates that DGA converges at the same rate as mirror descent for a simple problem and recovers the true mixture weights. To the best of our knowledge, this is the first theoretical convergence result for a gradient alignment descent.

## 3 EXPERIMENTS

Our experiments focus on two challenging cases. First, given *limited token resources* within each training domain, the model would risk overfitting with weights concentrated on a few domains. Second, given *large number of training domains*, applying DGA on domain reweighting could introduce intractable computation overheads linearly increasing according to the domain granularity.

**Generic Datasets and Domains.** For all the experiments, we use `Redpajama-v2` (Together AI Team, 2023) as the generic training set $D_{\text{gen}}$. This is one of the largest public corpus for LLM pretraining. `Redpajama-v2` contains 30 trillion filtered and deduplicated tokens from web-crawled dumps. Since this corpus does not come pre-segmented into domains, we obtain individual generic domains from $D_{\text{gen}}$ with clustering. Specifically, we use the embedding-and-clustering pipeline from Grangier et al. (2024). We first embed all the training sequences $\boldsymbol{x} \in D_{\text{gen}}$ with SentenceBert (all-MiniLM-L6-v2), yielding a 384 dimensional embedding $\text{Bert}(\boldsymbol{x})$. We then apply $k$-means clustering on the sentence embeddings into $k = 64$ clusters yielding $k$ domains $D_1, \ldots, D_k$.

To get fine-grained generic domains, we apply hierarchical clustering on the top of the first level of $k_1 = 64$ clusters. Specifically, each domain is further clustered once again into $64$ smaller clusters. We apply this strategy twice to get domains with granularity $k_2 = 64^2 = 4096$ and $k_3 = 64^3 = 262k$.

**Model Architecture.** We train small (125M), medium (350M) and large (750M) models with decoder-only transformers (Vaswani et al., 2017). We adopt most of the training settings and architectures from (Brown et al., 2020a). Their details are provided in Appendix C. For optimization, we use the AdamW optimizer (Loshchilov, 2017).

## 3.1 DOMAIN REWEIGHTING WITH LIMITED RESOURCES

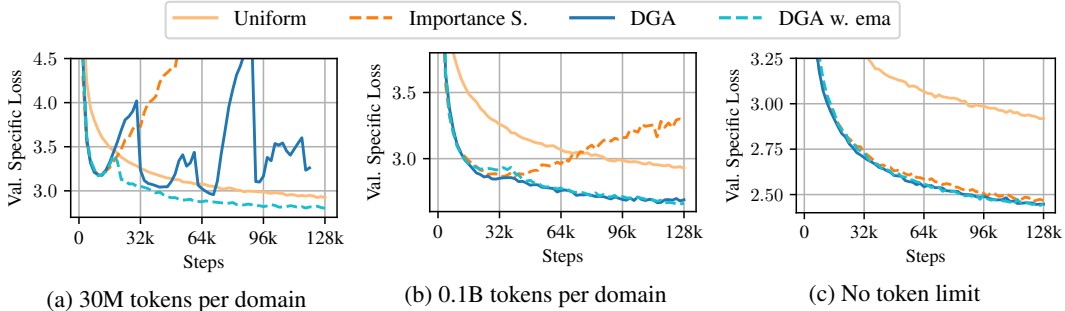

(a) 30M tokens per domain  (b) 0.1B tokens per domain  (c) No token limit

Figure 1: Comparing data reweighting methods with *free_law* as a specific set in a low generic data regime. When there are not enough tokens, importance sampling quickly overfits, while DGA manages to explore the training distributions to avoid overfitting. We see the importance of the EMA to stabilize DGA in the low data regime. When there is no token limit, adding an EMA ($\beta = 0.1$) does not negatively affect the performance.

Previous works on domain reweighting implicitly assume infinite token resources from all training domains (Xie et al., 2023a; Fan et al., 2024; Liu et al., 2024) while it is not always applicable in real-world cases. The scenario with limited training resources is challenging for online domain reweighting. Indeed, if the weights are concentrated on a few domains, e.g. on a single domain $D_i$, a large model will quickly overfit when the number of tokens in $D_i$ is small.

We expect DGA to mitigate overfitting by dynamically adjusting the domain weights. Specifically, once a model starts overfitting on $D_i$, the magnitude of the gradients $\nabla\ell(\boldsymbol{\theta}, D_i)$ decreases as its training loss $\ell(\boldsymbol{\theta}, D_i)$ is low, i.e. the domain knowledge from $D_i$ is well-learned. Consequently, the corresponding gradient alignment score $\boldsymbol{a}_i = \langle \nabla\ell(\boldsymbol{\theta}, D_i), \nabla\ell(\boldsymbol{\theta}, D_{\text{spe}}) \rangle$ decreases as well and DGA explores other domains with higher alignment scores. In other words, DGA down-weights domains once they are well-learned, thereby achieving a balance between ***exploration*** – by learning from diverse data domains – and ***exploitation***, by intensively training on the most relevant domains.

However, with limited data per domain, we remark that DGA without EMA demonstrates drastic changes at each domain weight update, focusing heavily on one domain at a time. Quickly changing domain weights is problematic since we want to use the same domain weights for $T_r$ steps in the future. This motivates the introduction of the EMA update in Algorithm 1, which regularizes the model and domain weights with the previous state when it starts to overfit.

**Experiment Setup.** We consider the generic domain split into $k = 64$ domains. We construct three scales of generic sets, either taking the full dataset or randomly sub-sampling 30M, 0.1B tokens per domain. For the targeted specific set $D_{\text{spe}}$, we use 5 subsets from *the Pile* (Gao et al., 2020) covering common specialized data types for LM applications: Math (*dm_mathematics*), Code (*github*, *stackexchange*), Medical (*pubmed_central*), Legal (*free_law*) and Scientific articles (*arxiv*).

We implement the importance sampling baseline described in subsection 2.3. We also compare to the *uniform baseline* with the domain weights $\boldsymbol{\alpha}_{\text{uniform}}$ as the natural proportion of each data domain in the generic Redpajama-v2 dataset. For importance sampling and uniform baselines, the domain weights are fixed throughout the entire training run. For both vanilla DGA and DGA with an EMA ($\beta = 0.1$), we update domain weights $\boldsymbol{\alpha}$ every $T_r = 100$ steps. We provide the ablation results on the step size $\eta$, frequency $T_r$ and ema factor $\beta$ in Appendix C. We use 125M models for experiments.

**Results.** We report the validation loss on the specialized set under various token constraints in Figure 1 for *free_law* and the results on other domains in Appendix A. With 30M tokens per domain, DGA with EMA effectively stabilizes the training, while vanilla DGA exhibits several loss spikes, suggesting a lack of robustness. Under a $0.1B$ token constraint, both DGA and DGA with EMA are able to dynamically adjust domain weights to mitigate overfitting. In contrast, fixed domain weights from importance sampling consistently lead to overfitting in token-limited scenarios, demonstrating the limitations of static weighting strategies in comparison to dynamic approaches like DGA. It is worth noting that adding the EMA has no negative effect on the learning efficacy when there is no token limit, which can be used as a robust regularization in the online domain reweighting context.

**Domain Weights Evolution.** In the experiments with a limited generic token budget (subsection 3.1), DGA without EMA often assigns excessive weight to one generic domain, leading to overfitting due to the restricted number of training tokens. This iterative over-weighting pattern on generic domain weights aligns with the observed loss spikes on the specific set (Figure 3a). In contrast, the EMA helps to regularize the weight dynamics, effectively preventing the model from overfitting by maintaining more balanced domain weights throughout the training process.

## 3.2 DISTRIBUTION REWEIGHTING: SCALING-UP DATA MIXING ON EXTREMELY FINE-GRAINED DATA DOMAINS

The computational overhead from DGA scales linearly with the number of domain $k$. This is intractable for datasets segmented in many fine-grained domains and, consequently, prior domain reweighting methods (Xie et al., 2023a; Fan et al., 2024; Liu et al., 2024; Kang et al., 2024) have not been applied in that setting. The fine-grained setting motivates *distribution reweighting* as an alternative to direct *domain reweighting*.

*Distribution reweighting* leverages the strength from both embedding-based (importance sampling) and gradient-based (DGA) strategies. We consider a generic training set partitioned into $k$ domains with a large $k$ (e.g. $4096, 262k$). We also have a set of $N$ auxiliary datasets $\{S_1, \ldots, S_N\}$, called *basis sets*, each from a specific domain of interest. We compute the importance sampling histograms for each basis set as $P = \{\boldsymbol{p}_1, \ldots, \boldsymbol{p}_N\}$, $\boldsymbol{p}_i \in \Delta^k, P \in R^{k \times N}$. We then use DGA to search over a reparameterized space leveraging this basis. We define the *domain weights* $\boldsymbol{\alpha}_{\text{domain}} \in \Delta^k$ as a convex combination of $N$ $k-$dimensional distributions derived from importance sampling,

$$\boldsymbol{\alpha}_{\text{domain}} \approx P\boldsymbol{\alpha}_{\text{dist}} = \alpha_{\text{dist},1} \cdot \boldsymbol{p}_1 + \alpha_{\text{dist},2} \cdot \boldsymbol{p}_2 + \ldots + \alpha_{\text{dist},N} \cdot \boldsymbol{p}_N \tag{8}$$

where the low-dimensional weights $\boldsymbol{\alpha}_{\text{dist}} \in \Delta^N$ are learned by DGA. This allows the use of fine-grained domain features while eliminating intensive gradient computation on each generic domain. Compared to the $(k+1)/T_r$ overheads from domain reweighting, applying distribution reweighting only incurs $(N+1)/T_r$ extra budget, where $N$ is typically much smaller than $k$. Importantly, this is equivalent to applying DGA with the $N$ generic domains $\tilde{D}_1, \ldots, \tilde{D}_N$ where $\tilde{D}_i = \text{mix}(\boldsymbol{p}_i)$. Hence, it does not require any modification to the base DGA algorithm; it suffices to be able to sample according to each $\text{mix}(\boldsymbol{p}_i)$. We provide the pseudo-code for the distribution reweighting with DGA in Appendix D.

**Experiment Setup.** We demonstrate the efficacy of distribution reweighting on the MMLU benchmark (Hendrycks et al., 2021). MMLU consists of 57 tasks from various knowledge fields, which serves as a testbed of multi-domain language modeling; by measuring the downstream accuracy, we can assess whether the improvements obtained in language modeling transfer to reasoning abilities.

We construct two specific datasets with different amounts of accessible samples: (1) `MMLU_a`: we take half of the examples from each task used as $D_{\text{spe}}$. We denote the other half of datapoints as `MMLU_b`, which is used for evaluation; (2) `MMLU_dev`: we randomly select 5 samples from each task, simulating the few-shot learning scenario. `MMLU_a` has $7.1k$ samples while `MMLU_dev` only has 285 samples, which yields sparse importance sampling histograms. For evaluation, we assess the language modeling performance by computing perplexity on `MMLU_b`. We also measure the accuracy for multiple choice question answering on `MMLU_b` with llm-eval (Gao et al., 2024).

We use generic domain splits with $k=64, 4096, 262k$ domains. We rely on 22 auxiliary sub-domains from *The Pile* (Gao et al., 2020) as our basis sets. For each auxiliary set, we take $15M$ tokens and compute their importance-sampling histograms as $\boldsymbol{p}_1, \ldots, \boldsymbol{p}_N \in \Delta^k$. To search for the optimal balance between diversity and specificity, we extend the basis

sets with the importance sampling histogram from the specific set itself (i.e. `MMLU_a` or `MMLU_dev`), yielding $N = 23$ distributions. For this experiment, we use 750M models.

**DGA greatly accelerates task-adaptive language modeling.** We evaluate the model's ability to acquire specialized knowledge on the target task (`MMLU`) in Appendix (Figure 12). DGA achieves substantial training speed-ups compared to uniform sampling, accelerating by $6.5\times$ on `MMLU_a` and $4.3\times$ on `MMLU_dev`. Moreover, with finergrained clustering ($k=262k$), DGA outperforms importance sampling, which suffers from sparse histograms and performance degradation, achieving $2\times$ and $3.2\times$ faster training on `MMLU_a` and `MMLU_dev`, respectively. These results highlight DGA's ability to effectively utilize finegrained domain information while avoiding overfitting, demonstrating its robustness in scenarios with limited specialized samples. We include details in subsection B.1 with the fine-tuning results in subsection G.1.

**Distribution reweighting makes better specialist and generalist.** While many task-adaptive algorithms are prone to catastrophic forgetting of general knowledge, we demonstrate that DGA with distribution reweighting effectively balances the acquisition of specialized knowledge while preserving general knowledge. To evaluate this, we report the loss on `MMLU_b`, representing specialized knowledge, alongside the average validation loss across 22 domains in *The Pile*, which reflects general knowledge drawn from broad and diverse domains (Figure 2). Compared to both importance sampling and uniform sampling baselines, DGA with distribution reweighting achieves a superior Pareto front, illustrating an improved trade-off between specialized performance and general knowledge.

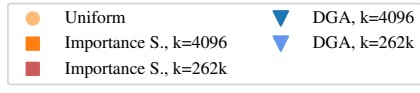

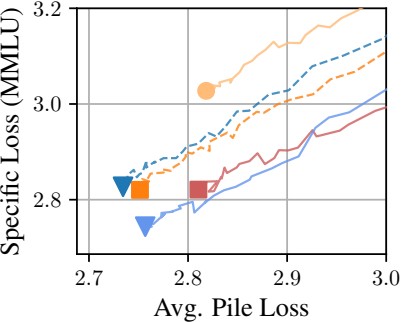

(a) `MMLU_a` (half the examples)

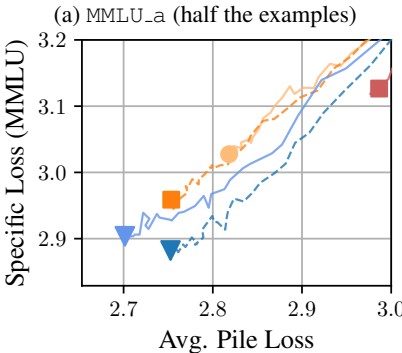

(b) `MMLU_dev` (5 examples per task)

Figure 2: Specialized and General Loss on Distribution Reweighting

## 4 DISCUSSION AND LIMITATIONS

**Comparison between *domain reweighting* and *distribution reweighting*.** As an efficient alternative of direct domain reweighting, we assess distribution reweighting in terms of specialized task adaptation and general domain losses. As shown in Figure 13, distribution reweighting outperforms domain reweighting in both specialized and general domain losses on $k=64$ clusters, while also incurring lower compute overhead. Furthermore, distribution reweighting exhibits remarkable scalability, with substantial improvements in specialized task perplexity as cluster granularity increases ($k=4096, 262k$). In contrast, domain reweighting struggles with scalability due to its high computational complexity, underscoring the efficiency and robustness of distribution reweighting in fine-grained settings.

**DGA outperforms DoGE on both specialized task adaptation and general knowledge.** We compare DGA with DoGE (Fan et al., 2024) in the context of task-adaptive pretraining. Comparing to both proxy model with online tuned domain weights and the base model with fixed optimized domain weights, DGA greatly outperforms DoGE in both specialized loss and general loss evaluated on the generic set. We present more details in Appendix F.

**Language modeling capability cannot be fully translated into reasoning accuracies.** According to Table 1, both importance sampling and DGA reweighting greatly outperform the uniform baseline on reasoning accuracy scores. However, despite the superior language modeling ability, DGA performs comparably as importance sampling in terms of accuracy scores. It indicates that better language modeling ability may not be fully transferable to better reasoning capabilities. We report the full results with different model scales in Appendix B with a detailed discussion on potential reasons.

**Weights Evolution on Distributions.** We present the evolution of domain weights for each basis distribution from DGA in Figure 3. Comparing different levels of granularity, with $k=262k$, the impor-

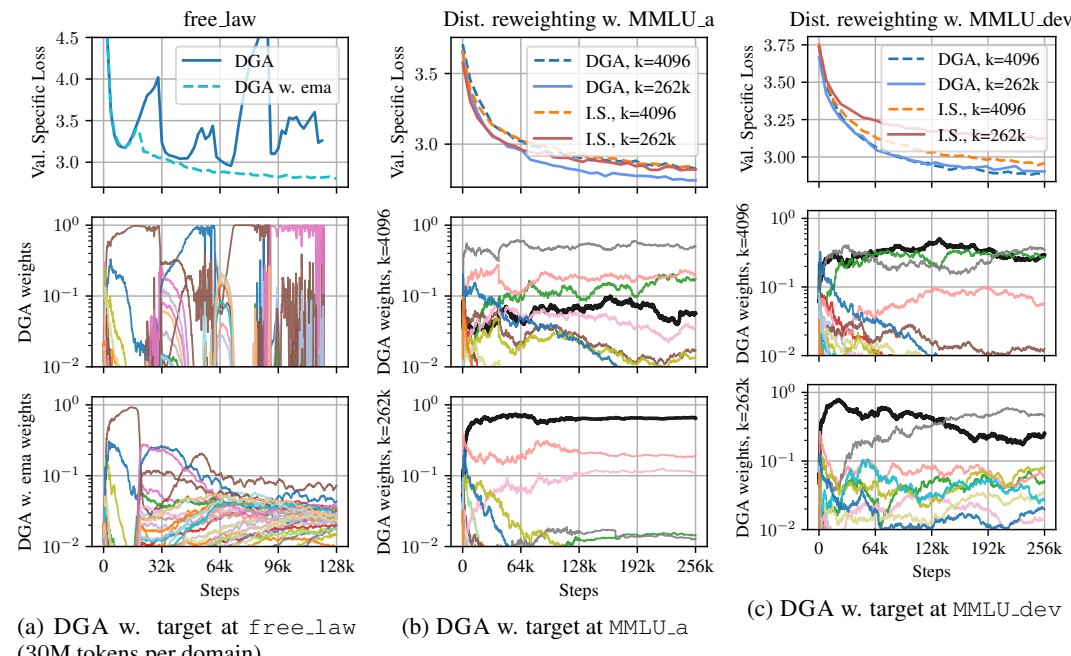

(a) DGA w. target at free_law (30M tokens per domain)

(b) DGA w. target at MMLU_a

(c) DGA w. target at MMLU_dev

Figure 3: The top row presents the specific loss over time, with the two bottom rows illustrating the evolution of domain (dist.) weights from DGA correspondingly, with each line representing a distinct domain. **Left**: Weights from the limited generic token experiment (subsection 3.1). **Middle** and **Right**: Weights from the distribution reweighting experiment (subsection 3.2). The thick black line highlights the dynamic weights assigned by DGA on the MMLU importance sampling distribution, which serves as a fixed training distribution for the importance sampling runs.

tance of the MMLU distribution is more emphasized than with $k$=4096, with the help of fine-grained domain features. Additionally, with sufficient samples from the specific domain (MMLU_a, Figure 3b), the MMLU distribution is consistently up-weighted across $262k$ generic domains. In contrast, on MMLU_dev, while the distribution on MMLU is initially up-weighted, it declines gradually in the late stage of training. Owing to the number of accessible samples from the specific set, the importance sampling distribution on MMLU_dev across $262k$ generic domains is very sparse. During the training, the learnability of the few activated generic domains diminishes, making other distributions more beneficial to the model.

In addition to the importance sampling distribution from the specific sets (MMLU_a and MMLU_dev), DGA effectively identifies other relevant distributions from The Pile that contribute to the learning on MMLU. These influential distributions, which include phil_papers, free_law, and dm_mathematics, are all considered to contain high-quality, academic-related contents. We present detailed curves with domain labels in subsection B.4. This ability to adaptively select beneficial distributions enhances the model's generalization and helps mitigate overfitting by leveraging a broader yet pertinent set of data sources during pretraining.

**Impact of generic domain granularity.** In Figure 4, we present the validation loss on the specific domain according to the number of clusters within the generic dataset. From $k = 64$ to $4096$, both DGA and importance sampling demonstrate significant improvement in language modeling in terms of validation loss (i.e., log of perplexity). However, when the number of clusters exceeds the scale of the accessible specific set, the importance sampling method overfits the limited number of activated generic domains, failing to capture broader domain knowledge. In contrast, DGA effectively leverages extremely fine-grained domain information across $262k$ generic domains with only $7k$ samples from MMLU_a. In the few-shot context (MMLU_dev), DGA mitigates a large performance degradation by utilizing diverse domain knowledge from other relevant distributions.

**Scaling performance of DGA on model scales.** To examine the scaling performance of the DGA algorithm, we train three models of varying scales (125M, 350M, and 750M) using both

uniform sampling and DGA on $k$=64 generic clusters. Model pretrained on DGA consistently outperforms uniform sampling baseline on specialized loss while exhibits degradation in general loss, i.e. validation loss on RedPajama-v2. We provide more details in subsection G.2.

## 5 RELATED WORK

**Task-adaptive Data Selection for Domain-Specific LLMs.** Many works have shown that one can effectively improve the LLM's performance on a specific downstream task with data selection according to the relevance of generic data for the targeted data domain. Gururangan et al. (2020) show that continued pretraining on data with high vocabulary overlap can boost its performance on the specific end-task. On machine translation task, Aharoni & Goldberg (2020) identify task-relevant pretraining datasets from a generic corpus using nearest neighbor of a small specialist dataset based on SentenceBert sentence representation. Wang et al. (2020); Grangier et al. (2023) train a small proxy model to give an importance weight per sample. Xie et al. (2023b) proposed DSIR as a lexical-based importance sampling method using n-gram features.

Other than feature-based importance sampling (Grangier et al., 2024), influence function-based method select data points which leads to the greatest loss drop on the target from the optimization perspective (Koh & Liang, 2020; Kwon et al., 2024; Agarwal et al., 2017). However, these methods often introduce intensive computational overheads from the second-order gradient computations, which is not applicable on large generic pretraining corpus.

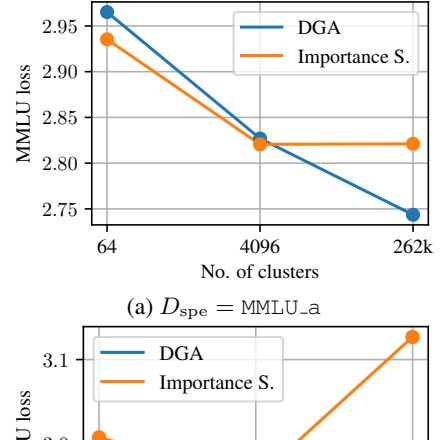

(a) $D_{\mathrm{spe}} = \texttt{MMLU\_a}$

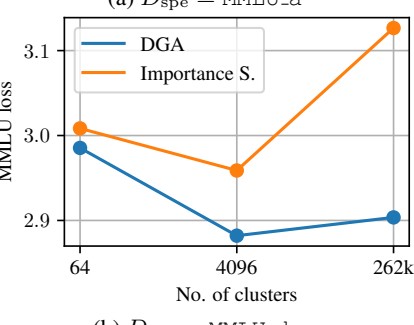

(b) $D_{\mathrm{spe}} = \texttt{MMLU\_dev}$

Figure 4: Impact of the generic set granularity for the distribution reweighting (subsection 3.2). We report the specific loss obtained after training for different granularities of the base clustering.

**Data Resampling through Domain Reweighting.** Given the large scale of the generic pretraining corpus, sample-level selection strategies are hard to implement for LLM pretraining. Alternatively, domain reweighting methods (Xie et al., 2023a; Fan et al., 2024; Liu et al., 2024; Kang et al., 2024) apply group-level selection by adjusting data sampling weights across different domains to reflect their importance. Based on the weak-to-strong generalization strategy (Burns et al., 2023), existing domain re-weighting methods typically estimate the optimal domain weights for a larger model based on the preferences of a small-scale proxy model. Xie et al. (2023a) apply group distributed robust optimization to optimize the worst-case loss gap between two small-scale proxies. Fan et al. (2024) use gradient alignment to dynamically adjust domain weights during proxy model training. Specifically, it identifies the most beneficial domains by aligning the gradients of the training data with the target task. However, it trains the proxy model on reweighted domain gradients to simulate the resampling scenario, which introduces more variance in the domain weights estimation.

## 6 CONCLUSION

To tackle two key challenges of online domain reweighting, we introduce Dynamic Gradient Alignment (DGA) as a stable and scalable data mixing method for LLM pretraining. Given a target task, DGA is an online algorithm that adjusts the training data distribution according to the current model status. This adaptation relies on an estimate of the progress on the target task from gradient alignments. We show that under limited tokens within generic domains, DGA with EMA can notably mitigate overfitting and yields superior performance on the end-task by balancing exploitation and exploration. We also propose a novel distribution reweighting strategy, which enables DGA to scale up to extremely fine-grained data domains without incurring intensive computations. Our experiments on MMLU show that applying distribution reweighting with DGA effectively leverages fine-grained domain knowledge to balance specialty and diversity during training. Our work demonstrates the scalability of gradient-alignment-based data reweighting methods, as well as their efficiency in data-constrained settings.

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

# A   TRAINING WITH LIMITED GENERIC TOKENS

## A.1   VALIDATION LOSS ON THE TARGETED END-TASK

We present the complete results on all six target domains (`arxiv`, `free_law`, `dm_mathematics`, `pubmed_central`, `github`, `stackexchange`) as follows. Across all six target domains, DGA with EMA ($\beta = 0.1$) consistently stablize the training and yields better language modelling performance under token-limited contexts.

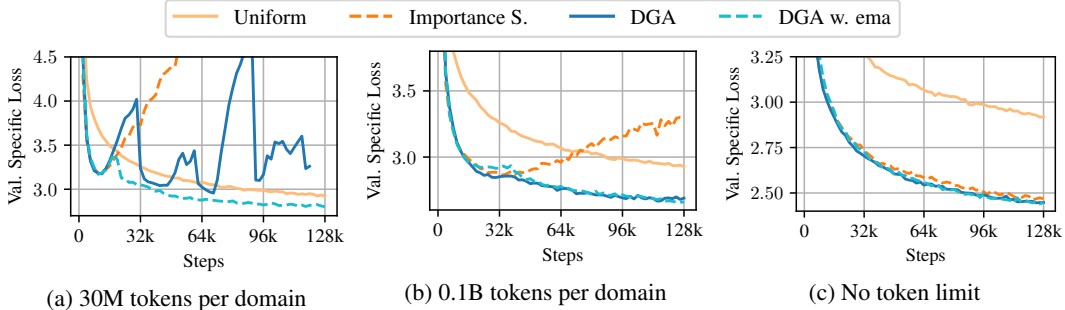

Figure 5: Results on all the domains for the low data experiment (subsection 3.1). The specific domain is `free_law`.

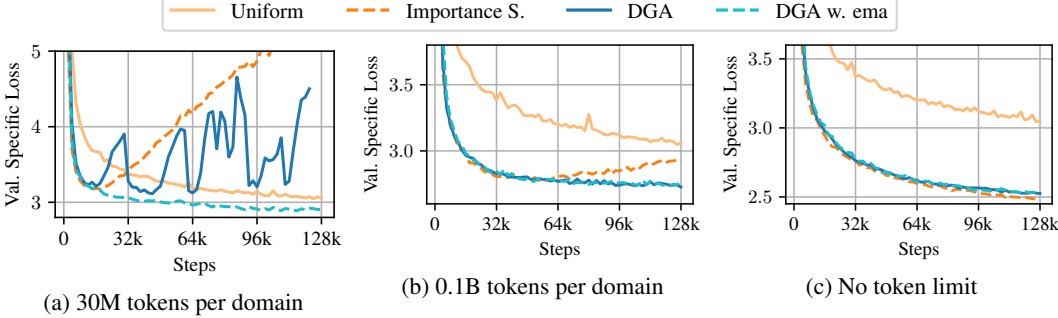

Figure 6: Results on all the domains for the low data experiment (subsection 3.1). The specific domain is `arxiv`

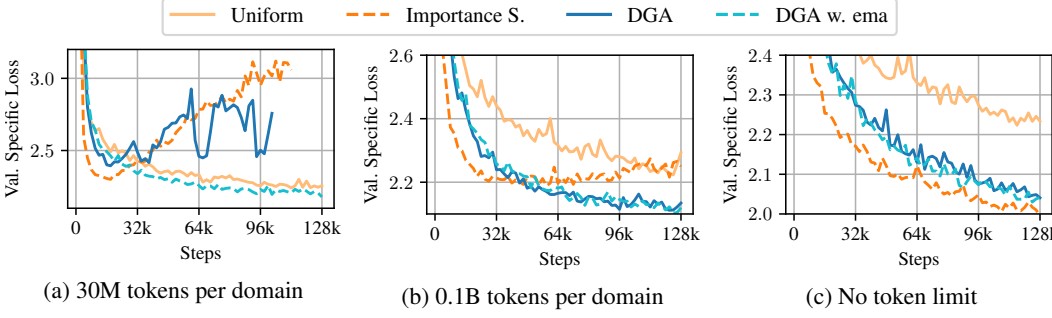

Figure 7: Results on all the domains for the low data experiment (subsection 3.1). The specific domain is `dm-mathematics`

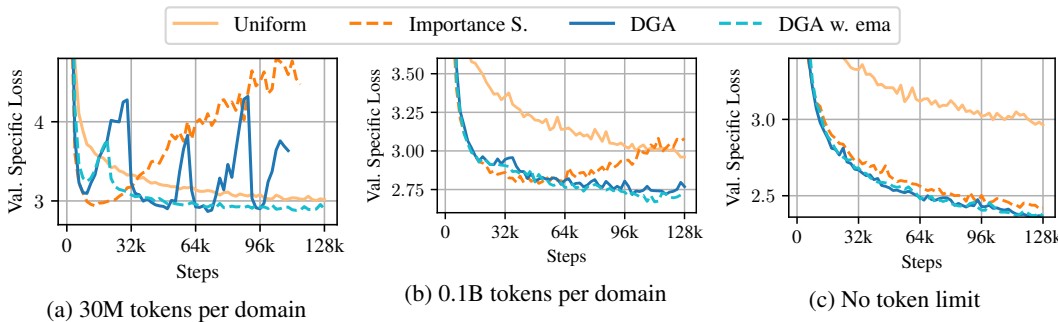

(a) 30M tokens per domain  (b) 0.1B tokens per domain  (c) No token limit

Figure 8: Results on all the domains for the low data experiment (subsection 3.1). The specific domain is github

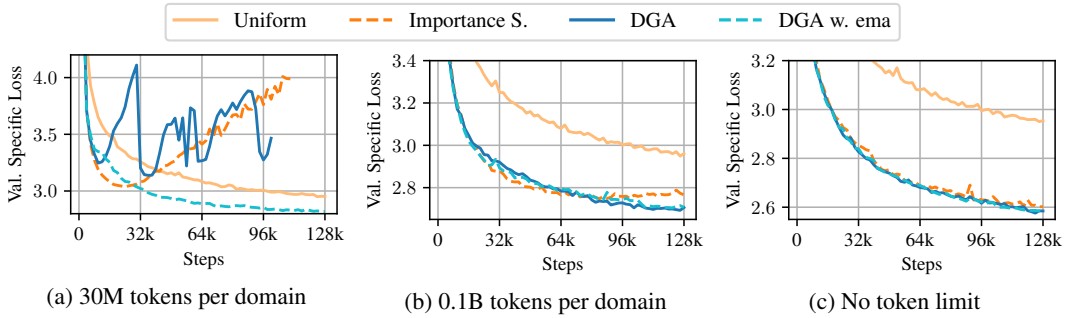

(a) 30M tokens per domain  (b) 0.1B tokens per domain  (c) No token limit

Figure 9: Results on all the domains for the low data experiment (subsection 3.1). The specific domain is pubmed-central

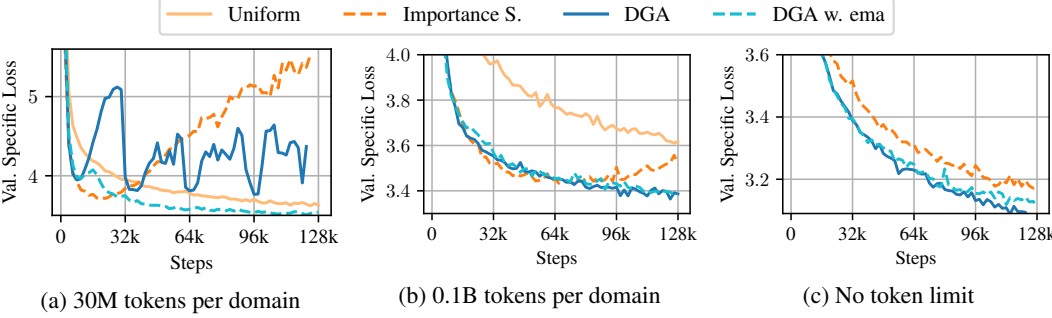

(a) 30M tokens per domain  (b) 0.1B tokens per domain  (c) No token limit

Figure 10: Results on all the domains for the low data experiment (subsection 3.1). The specific domain is stackexchange

## A.2 DOMAIN WEIGHTS EVOLUTION

We present the domain weights evolution on 64 generic domains from DGA with and w.o. EMA regularization. With both stackexchange and free_law as the specific set, EMA effectively smoothes the spiky domain weights, which therefore stablize the training process.

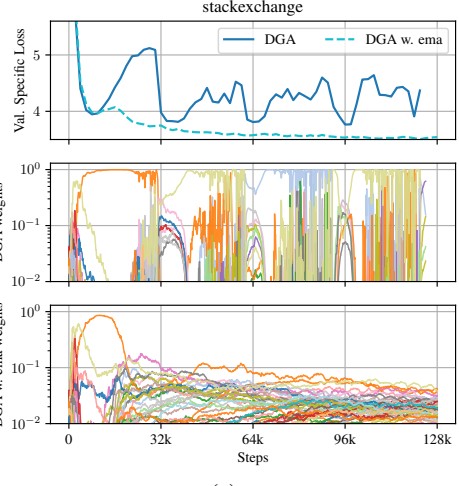 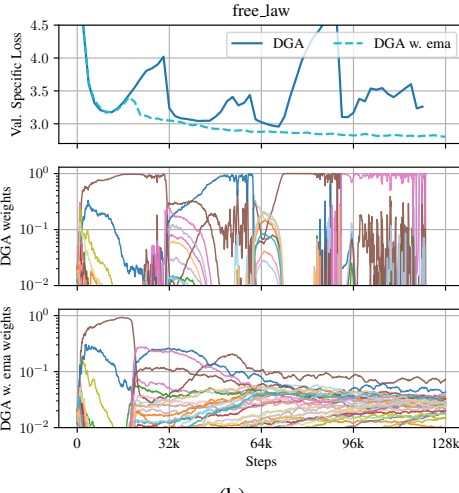

(a)                                        (b)

Figure 11: Comparing data reweighting methods with stackexchange (resp. free_law) as the specific set, in a low generic data regime. When there are not enough tokens, importance sampling quickly overfits, while DGA manages to explore the training distributions to avoid overfitting. We see the importance of the EMA to stabilize DGA in the low data regime.

# B  DISTRIBUTION REWEIGHTING

## B.1  DGA WITH DISTRIBUTION REWEIGHTING ACCELERATE TASK-ADAPTIVE TRAINING

We evaluate the perplexity on the target task (`MMLU`) in Figure 12, which reflects the model's ability to acquire specialized knowledge. Notably, DGA achieves significant acceleration compared to uniform sampling, with training speed-ups of $6.5\times$ and $4.3\times$ when targeting `MMLU_a` and `MMLU_dev`, respectively. Comparing to the importance sampling baseline, DGA achieves comparable performance when trained with `MMLU_a` on $k = 4096$ generic clusters. However, with specialized dataset as `MMLU_dev`, where samples from the specialized task are limited, DGA demonstrates a $2\times$ speed-up on $k = 4096$ clusters. Additionally, with a finer-grained generic clustering ($k = 262k$), DGA accelerates training by $2\times$ with `MMLU_a` and $3.2\times$ with `MMLU_dev`. The remarkable improvements highlight DGA's capability to effectively leverage fine-grained domain information and mitigate overfitting issues often seen with importance sampling with insufficient samples from specialized task.

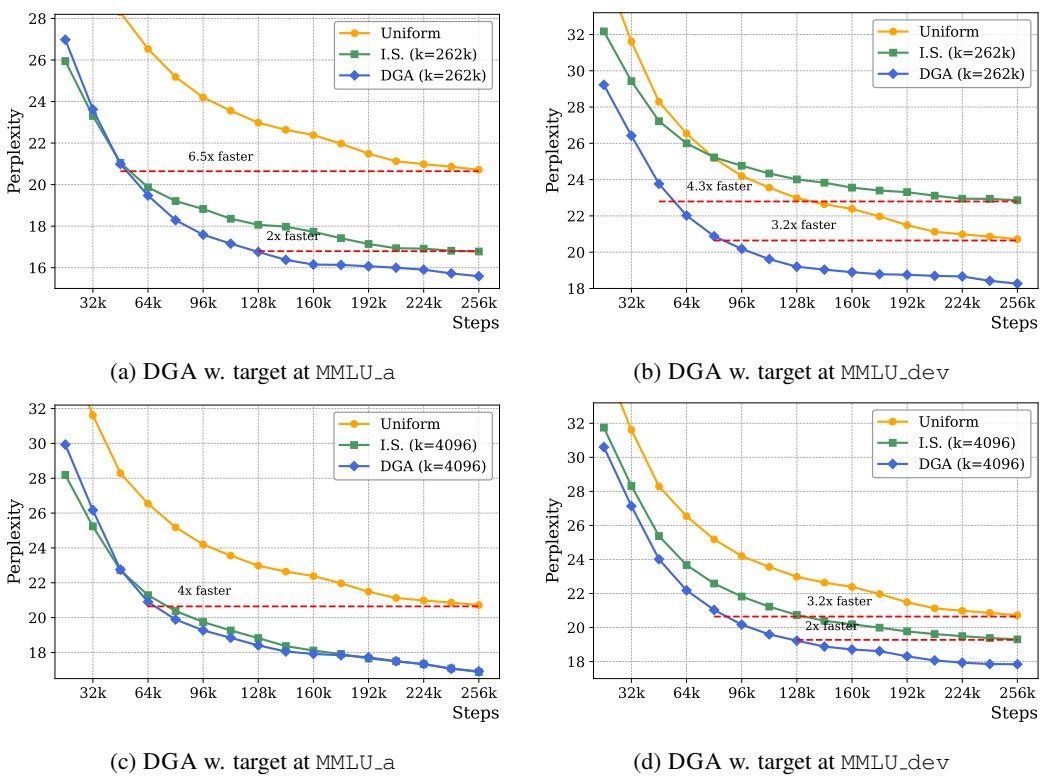

(a) DGA w. target at `MMLU_a`

(b) DGA w. target at `MMLU_dev`

(c) DGA w. target at `MMLU_a`

(d) DGA w. target at `MMLU_dev`

Figure 12: **Perplexity on the Specific Task (`MMLU`).** (a,c) present the specialized perplexity over time with target at `MMLU_a`; (b,d) present the results with target at `MMLU_dev`.

## B.2 Comparison between Domain Reweighting and Distribution Reweighting

In Figure 13, we compare distribution reweighting and domain reweighting according to the task-adaptive capability and the general knowledge preserved during task-adaptive training process. With distribution reweighting on $N = 23$ distributions and $k = 64$ domains, both specialized and generic perplexity are improved above domain reweighting on $k = 64$ domains. Meanwhile, the computational overheads are reduced from $k/T_r = 64\%$ to $N/T_r = 23\%$. The perplexity on the specialized domain can be significantly improved by increasing the domain granularity ($k = 64 \rightarrow 4096 \rightarrow 262k$), while the domain reweighting algorithm cannot scale up to large number of domains because of its high complexity.

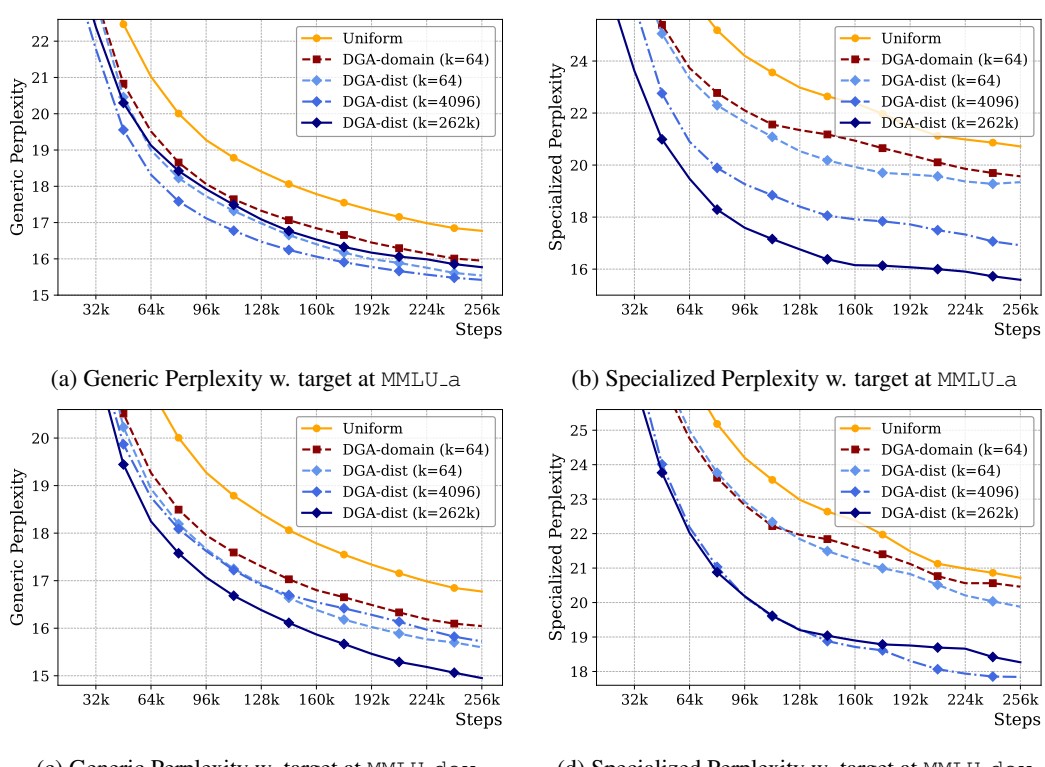

Figure 13: **Comparison between Domain Reweighting and Distribution Reweighting.** (a,c) present the perplexity across 22 domains in *The Pile*, which defined as the ***generic perplexity***; (b,d) present the ***specialized perplexity*** on MMLU.

## B.3 Evaluation Results on MMLU

### B.3.1 Reasoning Accuracy

Despite the superior performance on language modeling, DGA can hardly outperform importance sampling in terms of accuracy scores on MMLU benchmark (Table 1), which indicates the improvement on language modeling ability may not be able to fully translated to reasoning capacities. For the limited improvement on accuracy, we provide two potential explanations: (1) since the target objective in the outerloop ($L_{spe}$ in Equ. 3) is the next token prediction loss for language modeling, the language modeling performance is expected to be improved. However, it does not necessarily translate to the improved reasoning accuracy (2) MMLU is considered to be a challenging reasoning task, where the accuracies can hardly be improved when the model capacity is below some specific threshold. This phenomenon is well-illustrated by Figure 11. (G) in (Wei et al., 2022). How to derive an optimization objective which directly benefits the reasoning accuracy would be a compelling future direction.

Table 1: MMLU accuracies with domain reweighting methods. Both importance sampling and DGA reweighting greatly improve the accuracy above uniform baseline, while DGA does not show significant improvement above importance sampling.

| Method | | MMLU_a | MMLU_dev |
|---|---|---|---|
| Uniform | | 26.1 % | 26.1 % |
| Importance S. | $k = 4096$ | 27.7 % | 27.7 % |
| | $k = 262k$ | 28.4 % | 27.0 % |
| DGA dist. reweighting | $k = 4096$ | 26.8 % | 27.4 % |
| | $k = 262k$ | 28.0 % | 27.0 % |

### B.3.2  FULL EVALUATION RESULTS ACROSS DIFFERENT MODEL SCALES

We present the complete evaluation results on MMLU benchmark on small- (125M) and large-(750M) scale models. $k$ denotes the number of generic domains, $N$ denotes the number of reweighted importance sampling distributions from *basis sets*. $N$=22 indicates we only reweight 22 distributions from 22 *The Pile* subsets, while $N$=23 includes the importance sampling histgram from the specific set (MMLU). Since the 125M model shows marginal difference in accuracy because of limited capacity, we only scored 750M model on MMLU reasoning accuracies.

Table 2: Results on the domain reweighting experiment, with half MMLU as train set. The best results is **Bolded** and the second best is Underlined.

| 125M model | MMLU loss | MMLU acc. | avg. Pile loss |
|---|---|---|---|
| Uniform | 3.56 | - | 3.27 |
| Importance S. (k=4096) | 3.32 | - | 3.16 |
| Importance S. (k=262k) | **3.22** | - | 3.19 |
| DGA domain reweighting (k=64) | 3.31 | - | 3.10 |
| DGA dist. reweighting (N=22, k=4096) | 3.34 | - | 3.13 |
| DGA dist. reweighting (N=22, k=262k) | 3.34 | - | **3.05** |
| DGA dist. reweighting (N=23, k=4096) | 3.33 | - | 3.13 |
| DGA dist. reweighting (N=23, k=262k) | 3.25 | - | 3.10 |

| 750M model | MMLU loss | MMLU acc. | avg. Pile loss |
|---|---|---|---|
| Uniform | 3.03 | 26.1 % | 2.82 |
| Importance S. k=4096 | 2.82 | 27.7 % | 2.75 |
| Importance S. k=262k | 2.82 | **28.4** % | 2.81 |
| DGA domain reweighting (k=64) | 2.97 | 27.1 % | 2.77 |
| DGA dist. reweighting (N=22, k=4096) | 2.86 | 27.2 % | 2.73 |
| DGA dist. reweighting (N=22, k=262k) | 2.84 | 27.0 % | **2.66** |
| DGA dist. reweighting (N=23, k=4096) | 2.83 | 26.8 % | 2.73 |
| DGA dist. reweighting (N=23, k=262k) | **2.74** | 28.0 % | 2.76 |

Table 3: Results on the domain reweighting experiment, with 5 examples per task of MMLU as train set. We score only the 750M models.

| 125M model | MMLU loss | MMLU acc. | avg. Pile loss |
|---|---|---|---|
| Uniform | 3.56 | - | 3.27 |
| Importance S. (k=4096) | 3.40 | - | 3.16 |
| Importance S. (k=262k) | 3.41 | - | 3.29 |
| DGA domain reweighting (k=64) | 3.46 | - | 3.19 |
| DGA dist. reweighting (N=22, k=4096) | 3.37 | - | 3.12 |
| DGA dist. reweighting (N=22, k=262k) | 3.36 | - | **3.03** |
| DGA dist. reweighting (N=23, k=4096) | 3.37 | - | 3.14 |
| DGA dist. reweighting (N=23, k=262k) | **3.35** | - | 3.08 |

| 750M model | MMLU loss | MMLU acc. | avg. Pile loss |
|---|---|---|---|
| Uniform | 3.03 | 26.1 % | 2.82 |
| Importance S. k=4096 | 2.96 | 27.7 % | 2.75 |
| Importance S. k=262k | 3.13 | 27.0 % | 2.99 |
| DGA domain reweighting (k=64) | 3.01 | 27.0 % | 2.77 |
| DGA dist. reweighting (N=22, k=4096) | 2.89 | 26.8 % | 2.76 |
| DGA dist. reweighting (N=22, k=262k) | 2.93 | 27.0 % | **2.68** |
| DGA dist. reweighting (N=23, k=4096) | **2.88** | **27.4** % | 2.75 |
| DGA dist. reweighting (N=23, k=262k) | 2.90 | 27.0 % | 2.70 |

## B.4 WEIGHTS EVOLUTION ON DISTRIBUTIONS

We present the weights assigned to each distribution over time corresponding to the loss on the specialized task in Figure 14. The top-10 up-weighted distributions include one from the specific dataset $D_{spe}$ and other academic related domains (e.g. pubmed_central, phil_papers, dm_mathematics), which are greatly relevant to the sub-topics in MMLU benchmark.

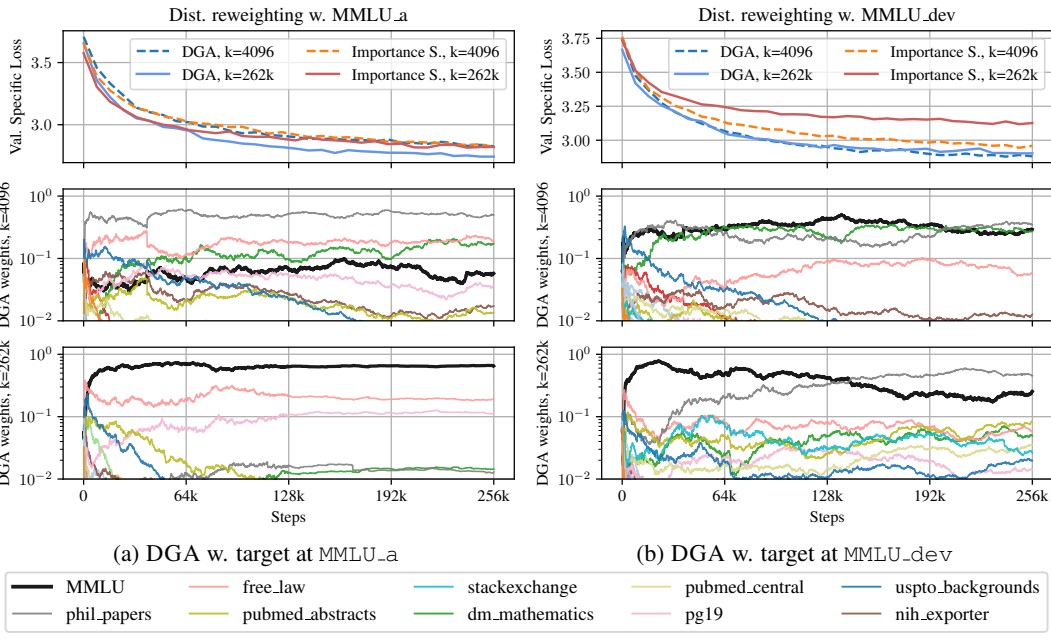

(a) DGA w. target at MMLU_a

(b) DGA w. target at MMLU_dev

(c) Top 10 upweighted distributions

Figure 14: The top row presents the specific loss over time, with the two bottom rows illustrating the evolution of domain (dist.) weights from DGA correspondingly, with each line representing a distinct domain.

## C  HYPERPARAMETERS

Table 4 provides the model architectures and hyperparameters used in this paper.

Table 4: Architecture hyperparameters for various model scales used in the paper. All models are vanilla Transformer decoder-only models.

|      | Layers | Attention heads | Embed dim | Hidden dim | Context limit | learning rate |
|------|--------|-----------------|-----------|------------|---------------|---------------|
| 125M | 12     | 12              | 768       | 3072       | 1024          | $1 \times 10^{-4}$ |
| 350M | 24     | 16              | 1024      | 4096       | 1024          | $1 \times 10^{-4}$ |
| 750M | 36     | 20              | 1280      | 5120       | 1024          | $1 \times 10^{-4}$ |

## D  DGA FOR DISTRIBUTION REWEIGHTING

Algorithm 2 explains the distribution reweighting with DGA. The implementation can be easily adapted from domain reweighting DGA with minor modifications.

---

**Algorithm 2** Distribution Reweighting w. DGA. (Difference from domain reweighting are marked in blue)

---

1: **Input:** Generic domains $D_1, \ldots, D_k$, I.S. distributions $\mathcal{A}_{dist} \triangleq [\boldsymbol{p}_1, \ldots, \boldsymbol{p}_N]$, specific set $D_{\text{spe}}$, inner optimizer state $\boldsymbol{\omega}^0$, optimizer function $\texttt{Optimizer}$ such as Adam or SGD, initial weights $\boldsymbol{\alpha}^0$, outer learning rate $\eta$, weight update frequency $T_r$

2: **Initialize distribution weights**: $\boldsymbol{\alpha}^0_{\text{dist}} = \boldsymbol{\alpha}^0$, i.e. init. domain weights: $\boldsymbol{\alpha}^0_{\text{domain}} = \boldsymbol{\alpha}^0_{\text{dist}} \otimes \mathcal{A}_{dist}$.

3: **for** $t = 0 \ldots T$ **do**

4:     Sample batch from generic mixture: $\boldsymbol{x} \sim \text{mix}(\boldsymbol{\alpha}^t_{\text{domain}})$

5:     Update the parameters $\boldsymbol{\theta}^{t+1}, \boldsymbol{\omega}^{t+1} \leftarrow \texttt{Optimizer}(\boldsymbol{\theta}^t, \boldsymbol{\omega}^t, \nabla_{\boldsymbol{\theta}} \ell(\boldsymbol{\theta}^t, \boldsymbol{x}))$

6:     **if** $t \% T_r = 0$ **then**

7:         Sample a batch from each *distribution*: $\boldsymbol{x}_i \sim \text{mix}(\boldsymbol{p}_i)$ for $i = 1 \ldots N$ and $\boldsymbol{y} \sim D_{\text{spe}}$

8:         Compute gradient alignements $\boldsymbol{a}^t_i \leftarrow \langle \nabla \ell(\boldsymbol{\theta}^{t+1}, \boldsymbol{x}_i), \nabla \ell'(\boldsymbol{\theta}^{t+1}, \boldsymbol{y}) \rangle$

9:         Update *distribution weights*: $\boldsymbol{\alpha}^{t+1}_{\text{dist}} \leftarrow \frac{\hat{\boldsymbol{\alpha}}}{\sum_{i=1}^k \hat{\boldsymbol{\alpha}}_i}$ with $\hat{\boldsymbol{\alpha}} = \boldsymbol{\alpha}^t_{\text{dist}} \odot \exp(-\eta \boldsymbol{a}^t)$,

10:        Updated *domain weights*: $\boldsymbol{\alpha}^{t+1}_{\text{domain}} = \boldsymbol{\alpha}^{t+1}_{\text{dist}} \otimes \mathcal{A}_{dist}$.

11:    **else**

12:        Do nothing: $\boldsymbol{\alpha}^{t+1}_{\text{dist}} \leftarrow \boldsymbol{\alpha}^t_{\text{dist}}$

13:    **end if**

14: **end for**

15: **Return** Optimized parameters $\boldsymbol{\theta}^{(T)}$ and weights trajectory $\boldsymbol{\alpha}^t, t = 0 \ldots T$

---

# E  PROOF OF THEOREM 2

To ease notations, we define $m(\boldsymbol{\alpha}) = \sum_{i=1}^{k} \alpha_i \boldsymbol{\mu}_i$. Let $\boldsymbol{\alpha}^t$ be the current weight estimate of DGA and $\boldsymbol{\theta}^t$ be the estimate of the parameters. The generalist gradient is

$$\nabla_{\boldsymbol{\theta}} L(\boldsymbol{\theta}^t, \boldsymbol{\alpha}^t) = \sum_{i=1}^{k} \alpha_i(\boldsymbol{\theta}^t - \boldsymbol{\mu}_i)$$

Hence, doing a gradient descent step on $\boldsymbol{\theta}^t$ with step 1 yields

$$\boldsymbol{\theta}^{t+1} = \boldsymbol{\theta}^t - \nabla_{\boldsymbol{\theta}} L(\boldsymbol{\theta}^t, \boldsymbol{\alpha}^t) \tag{9}$$

$$= \sum_{i=1}^{k} \alpha_i \boldsymbol{\mu}_i = m(\boldsymbol{\alpha}) \tag{10}$$

Then, the gradient alignment is

$$a_i = \langle \nabla_{\boldsymbol{\theta}} L_i(\boldsymbol{\theta}^{t+1}), \nabla_{\boldsymbol{\theta}} L_{\mathrm{spe}}(\boldsymbol{\theta}^{t+1}) \tag{11}$$

$$= \langle m(\boldsymbol{\alpha}^t) - \boldsymbol{\mu}_i, m(\boldsymbol{\alpha}) - m(\tilde{\boldsymbol{\alpha}}) \rangle] \tag{12}$$

$$= \langle m(\boldsymbol{\alpha}^t), m(\boldsymbol{\alpha}) - m(\tilde{\boldsymbol{\alpha}}) \rangle - \Sigma_{j=1}^{k} \langle \boldsymbol{\mu}_i, \boldsymbol{\mu}_j \rangle (\boldsymbol{\alpha}^t - \tilde{\boldsymbol{\alpha}}) \tag{13}$$

$$= \langle m(\boldsymbol{\alpha}^t), m(\boldsymbol{\alpha}) - m(\tilde{\boldsymbol{\alpha}}) \rangle - \left[ MM^T(\boldsymbol{\alpha}^t - \tilde{\boldsymbol{\alpha}}) \right] \tag{14}$$

The first part does not depend on $i$, so it will have no effect in the mirror descent step. Hence, the mirror descent step is equivalent to a mirror descent step to minimize the function

$$f(\boldsymbol{\alpha}) = \frac{1}{2} \| M(\boldsymbol{\alpha} - \tilde{\boldsymbol{\alpha}}) \|^2 \ .$$

The convergence theory of mirror descent yields the bound (Beck & Teboulle, 2003, Theorem 4.2)

$$\min_{t=1...T} f(\boldsymbol{\alpha}^t) = O(\frac{1}{\sqrt{T}})$$

which in turn, thanks to the strong-convexity of $f$, gives

$$\min_{t=1...T} \| \boldsymbol{\alpha}^t - \tilde{\boldsymbol{\alpha}} \|^2 \leq \frac{1}{\lambda_{min}(MM^T)} \min_{t=1...T} f(\boldsymbol{\alpha}^t) = O(\frac{1}{\sqrt{T}})$$

# F  COMPARISON BETWEEN DGA AND DOGE

We compare DGA with DoGE (Fan et al., 2024) in the context of task-adaptive training. Specifically, we use the `RedPajama-v2` dataset clustered into $k$=64 domains as the generic dataset, and `Arxiv` from *The Pile* as the specialized domain.

Following the methodology outlined by Fan et al. (2024), we implement a two-stage pipeline for domain-reweighted pretraining. First, we optimize the domain weights $\alpha$ on a proxy model. Next, we compute the average domain weights over all time steps and use them as fixed sampling weights to train the base model. For this setup, we train proxy models at two scales (31M and 125M parameters) and a base model at 125M parameters. For DGA, we follow Algorithm 1 to train the 125M base model. Unlike DoGE, DGA updates the domain weights dynamically in an online manner, eliminating the need for a separate proxy model or precomputed fixed sampling weights.

We report both the specialized loss on `Arxiv` over time (measured in GPT hours) and the validation loss on `RedPajama-v2`, which reflects the retention of general knowledge from the generic dataset. To evaluate the performance of DoGE as both an online and offline domain reweighting method, we present results from both the proxy model and the base model. All the models are trained with $1 \times$ Nvidia A100 GPU.

According to Figure 15 (a), DGA greatly outperforms online DoGE (125M proxy model) and achieves comparable specialized loss as the offline DoGE (125M model trained w. 31M/125M Doge weights). However, as an online algorithm, DGA does not require any proxies, which significantly outperform DoGE in terms of efficiency. In addition, as shown in Figure 15 (b), DGA reaches a lower loss on RedPajama than both online and offline DoGE. It indicates that DoGE is more vulnerable from catastrophic forgetting compared to DGA.

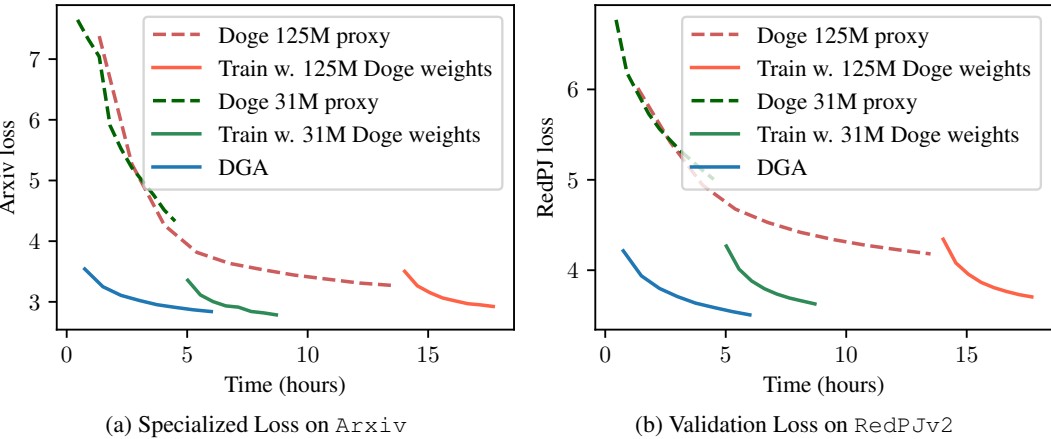

(a) Specialized Loss on `Arxiv`      (b) Validation Loss on `RedPJv2`

Figure 15: **Comparison between DGA and DoGE on Language Modeling**

# G    SUPPLEMENT RESULTS ON LANGUAGE MODELING

## G.1    TASK-ADAPTIVE PRETRAINING WITH FINE-TUNING

We further assess the model's performance after fine-tuning on a dataset drawn from the downstream task. Specifically, we fine-tune 125M pretrained model checkpoints trained using uniform sampling and DGA reweighting with $k$=64, on tokens from the `Stackexchange` subset of *The Pile*. We chose different numbers of tokens available for fine-tuning. For each number of tokens, we train the model on those tokens only using a small learning rate ($10^{-5}$), and report the best validation loss across the runs.

As shown in Figure 16, the model pretrained with DGA demonstrates superior performance on the specialized domain (`Stackexchange`) after fine-tuning, consistently outperforming the uniform sampling and importance sampling baselines across various scales of available fine-tuning tokens.

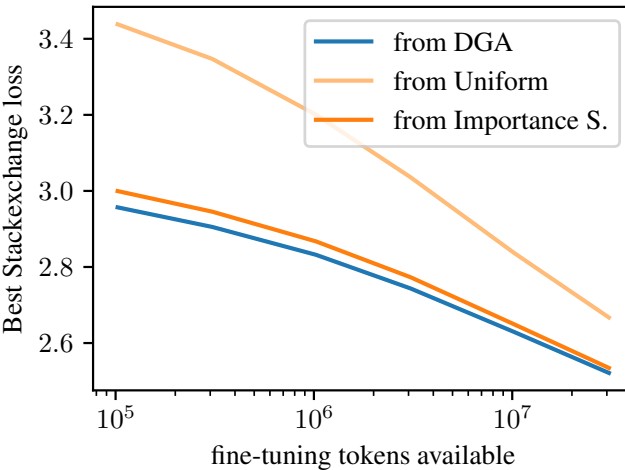

Figure 16: **Specialized Loss on `Stackexchange` after Task-specific Fine-tuning.**

## G.2 Scaling Performance of DGA across Various Model Scales

To examine the scaling performance of the DGA algorithm, we train three models of varying scales (125M, 350M, and 750M) using both uniform sampling and DGA with $k$=64 clusters. As illustrated in Figure 19 (a), DGA consistently outperforms uniform sampling by a significant margin across all three model scales. However, as shown in Figure 19 (b), DGA exhibits some degradation in general knowledge compared to uniform sampling, which presents the highest level of sample diversity. Notably, this performance gap in general knowledge narrows on the largest model (750M), highlighting the potential scalability benefits of DGA in larger model regimes.

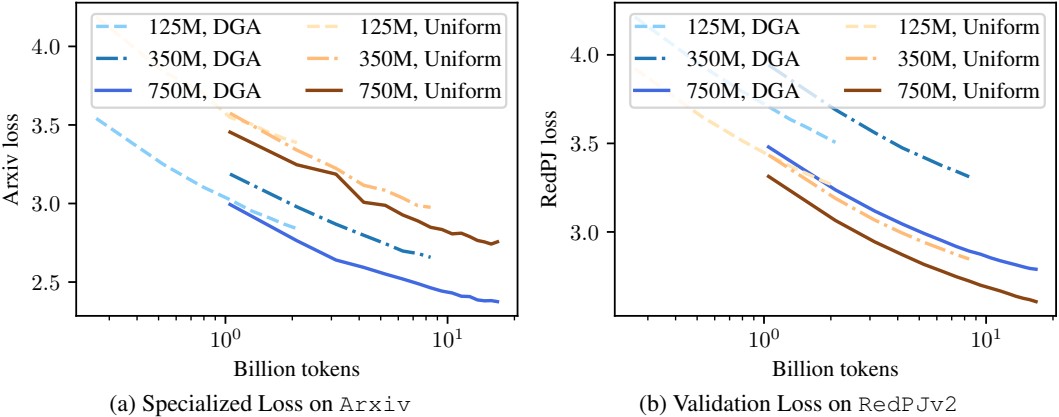

(a) Specialized Loss on `Arxiv`   (b) Validation Loss on `RedPJv2`

Figure 17: **Language Modeling Performance across Model Scales**

## G.3 ABLATION ON HYPER-PARAMETERS

**Ablations on step size $\eta$ and update frequency $T_r$.**   We conducted the ablation experiments on both step size $\eta$ and frequency $T_r$ used for updating domain weights $\alpha$, as described in Algorithm 1. According to Figure 18, using a step size that is too small will result in slow updates to the domain weights. On the other hand, applying a large step size ($\eta$) can accelerate the learning of domain weights but may also lead to training instability due to overly up-weighted domains. Since EMA can effectively stabilize learning, we recommend that practitioners choose $\eta$ values between 0.1 and 0.5.

Regarding the reweight frequency, using a smaller $T_r$ generally improves the final specialized loss but increases computation costs. To balance cost and performance, we recommend setting $T_r$ between 30 and 100. However, these values should be tailored to specific use cases and levels of domain granularity.

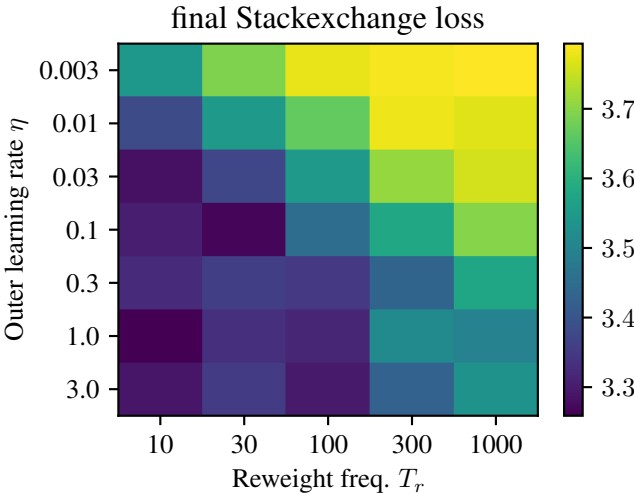

Figure 18: **Grid search on step size $\eta$ and update frequency $T_r$**

**Ablations on EMA factor** $\beta$. We perform an ablation on the hyper-parameter $\beta$ applied in the exponential moving average update in Algorithm 1. We choose two extreme values: (1) $\beta$=0.1, which is close to 0 and (2) $\beta$=0.9, which is close to 1.0. We present the specialized loss on various target domains over time with different values of $\beta$. With a strict token limit (30M per domain), both $\beta$=0.1, 0.9 can effectively smooth the loss curve while efficiently acquiring specialized knowledge. However, in the cases with 0.1B token per domain or unlimited resources, setting $\beta$=0.9 severely slow down the learning of domain weights, which hurts training efficiency.

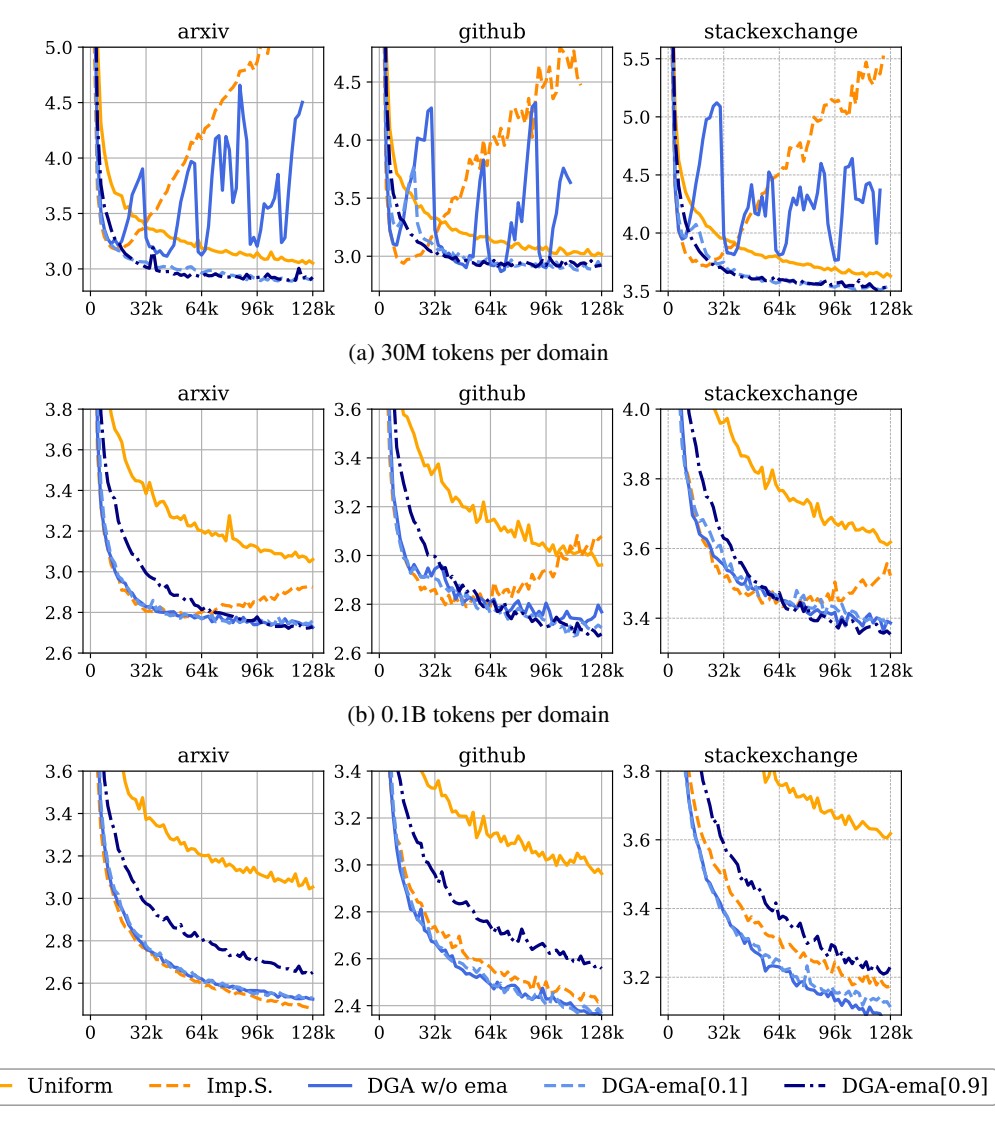

Figure 19: **Language Modeling Performance across Model Scales**

# H  CATASTROPHIC FORGETTING ON GENERAL KNOWLEDGE

As a prevalent issue in task-adaptive training, catastrophic forgetting often arises when models lose general knowledge while adapting to specialized tasks. Our experimental results demonstrate that DGA can preserve more general knowledge during task-adaptive pretraining, while other baselines, including importance sampling (Figure 2) and DoGE (Figure 15) are more vulnerable to catastrophic forgetting.

In addition, compared with direct domain reweighting, our proposed distribution reweighting algorithm can retain the general domain knowledge better without sacrificing specialized loss (Figure 13).

