# OpenReview forum: "Dynamic Gradient Alignment for Online Data Mixing"
_ICLR.cc/2025/Conference — Submitted to ICLR 2025_

### Official Review · Reviewer_9LhJ · 2024-11-02

**Soundness:** 3
**Presentation:** 3
**Contribution:** 3
**Rating:** 5
**Confidence:** 3

**Summary:**

The paper introduces Dynamic Gradient Alignment (DGA), an online data reweighting algorithm aimed at optimizing data mixtures for training large language models (LLMs) on specific tasks. Unlike static or previously established gradient-alignment methods, DGA dynamically updates domain weights during training by aligning model gradients on the target task, leading to improved performance and reduced overfitting, especially in data-constrained environments. The method is scalable, incurs minimal computational overhead, and leverages exponential moving averages (EMA) for stabilizing updates.

**Strengths:**

- **Innovative Approach**: The paper presents DGA as an online, scalable alternative to existing domain reweighting methods, offering significant advantages in flexibility and adaptability.
- **Empirical Evidence**: Detailed experiments show DGA's effectiveness compared to importance sampling, particularly under token-constrained conditions and fine-grained data domains.
- **Theoretical Rationale**: The method is well-anchored in optimization principles, connecting gradient alignment and bilevel optimization frameworks.
- **Practical Relevance**: DGA demonstrates strong utility for real-world data scenarios with limited resources, showcasing improved balance between exploration and exploitation in domain weighting.

**Weaknesses:**

- **Theoretical Assumptions**: While the approach connects well to optimization theories, the convergence proofs are limited due to non-convexity in practice.
- **Specificity vs. Generality**: The results on downstream reasoning (MMLU) indicate that while DGA improves language modeling, the gains do not always translate to enhanced reasoning capabilities, suggesting potential limits in generalizability.
- **Sparse Baseline Comparisons**: The paper could benefit from more varied baseline methods beyond standard importance sampling to contextualize DGA's performance.
- **Stability Under Extreme Data Constraints**: Although EMA helps with stability, the paper notes challenges when DGA operates without it, implying potential reliability issues under certain configurations.

**Questions:**

- Can the authors provide insights on the performance of DGA on tasks beyond language modeling to substantiate its generalization claims?
- What are the primary factors influencing DGA's stability when EMA is not used?
- Are there any plans to explore hybrid approaches combining DGA with more computationally intensive, sample-level methods?

---

> ### Author Response · Authors · 2024-11-22
> **Author response to reviewer 9LhJ**
>
> We thank the reviewer for the feedback and insightful suggestions for improving our work! We provide corresponding clarifications as follows:
>
> ## W1: Proof of convergence
> As the reviewer points out, the theoretical understanding of data mixing (i.e. domain reweighting) algorithms is a valuable topic that could significantly enhance the theoretical contributions of our paper.  We have updated the paper to include a convergence proof of DGA in a simple convex scenario (**Theorem 2**). To the best of our knowledge, this is the first theoretical proof of convergence for a data reweighting algorithm. For instance, prior works such as [1,2] do not prove the convergence of their algorithms.
>
> ## W2: Potential explanations for the limited improvement in reasoning accuracy
> We agree with the reviewer that despite a substantial improvement above the uniform baseline, DGA performs comparably importance sampling baselines in reasoning accuracy scores, as shown in Table 1.
>
> As for the considerable gap between language modeling ability and reasoning accuracy, we provide two potential explanations in **Appendix B.3.1**: (1) since the target objective in the outer-loop ($L_{spe}$ in Equ. 3) is the next token prediction loss for language modeling, the language modeling performance is expected to be improved. However, it does not necessarily translate to improved reasoning accuracy (2) MMLU is considered to be a challenging reasoning task, where the accuracies can hardly be improved when the model capacity is below some specific threshold (Figure 11. (G) in [3]). Understanding the loss-accuracy gap and deriving an optimization objective that directly benefits the reasoning capability would be a compelling future direction.
>
> ## W3: Comparisons to more baselines
> We have updated the paper and included the comparison between DGA and DoGE in Appendix F (Fig. 15). The results demonstrate that DGA outperforms DoGE in both specialized loss and general loss.
>
> ## Q1: Insights on other tasks
> In this work, we mainly focus on the language modeling tasks. However, we think this gradient-based method could be generally applied to various tasks, e.g. image classification and vision-language modeling. Since the preprocessing (embedding and clustering) of the pretraining corpus takes intensive effort, we leave more broad benchmarking as one future extension of our current work.
>
> ## W4&Q2: What factors influence DGA's stability when EMA is not used?
> According to the domain weight evolution trajectories shown in Figure 3(a), DGA without EMA consistently up-weights a specific domain for a certain number of steps before shifting to another. This cyclical behavior amplifies overfitting over time and optimizes the model parameters to an ill-positioned location, which could be fatal for the online optimization. Also, the first identified relevant domain to be significantly down-weighted when the model shifts to the second domain.
> Notably, the spikes in domain weights align closely with those observed in the validation loss curves, highlighting the correlation between these dynamics and the model's performance fluctuations.
>
>
> ## Q3: Future directions on sample-level data selection
> While the sample-level selection is not computationally friendly on pretraining because of the gigantic scale of the pretraining corpus, the gradient-based method could be applied to sample-level data selection in continual pretraining or task-specific finetuning stages. Additionally, the clustering-based method could be applied to balance the diversity and relevance during data selection. We consider it as a promising direction for future research.
>
> We hope our answers can resolve most of the concerns. If you have any further questions, don’t hesitate to contact us. We will improve the manuscript according to your valuable suggestions!
>
> [1] Doge: Domain reweighting with generalization estimation [2] Doremi: Optimizing data mixtures speeds up language model pretraining [3] Emergent Abilities of Large Language Models

---

> ### Author Response · Authors · 2024-12-02
> **End of the discussion period**
>
> Dear reviewer 9LhJ,
>
> Thank you again for your efforts reviewing our paper and providing valuable feedback! Since **tomorrow is the last day of the author-reviewer discussion phase**, we hope you can take some time to look into our rebuttals and additional results. Let us know if you have other concerns and questions!
>
> Sincerely,
>
> the authors

---

### Official Review · Reviewer_eJDq · 2024-11-04

**Soundness:** 3
**Presentation:** 3
**Contribution:** 2
**Rating:** 6
**Confidence:** 4

**Summary:**

The submission proposes a new method to improve training of LLMs on datasets that have an increased diversity / size ratio, e.g. mix of many small datasets. The method is a combination of increased (local) adaptability via dynamic gradient alignment (DGA) and smoothing via a moving average (EMA). Results report effects on training loss and accuracy in comparison to prior art of "Importance Sampling" with the same target.

**Strengths:**

The submission is well-written and a good mixture of motivation, analysis of previous work, detailing theoretic foundations, experimental verification and conclusions.

**Weaknesses:**

Experimental results only show loss, but no accuracy improvements over Importance Sampling (Table 1). Tables 2 & 3 also show no consistency in the pattern of first and second best of the compared approaches and alternating "wins" by tiny margins. This suggests that the proposed technique is just possibly only an alternative to Importance Sampling.

Initially, I had some issues correctly understanding the term "online" in the name of the proposed technique. Similar to me it may mislead other readers to think that this postpones the mixing to inference time (that would be an awesome new result), but it clearly doesn't.

A slight weakness lies in the fact that the technique is based on the assumption of LLMs being trained via gradients. This limits applicability to the current state-of-the-art. In contrast, most sampling techniques are more generally applicable.

**Questions:**

Lines 067-069 ("Since the domain weights... leads to snow-balled errors."): this claim is vague and should be verified by experiment(s). Or did I miss that (maybe Figure 1(a)? Can you please add clarity to this?

---

> ### Author Response · Authors · 2024-11-22
> **Author response to reviewer eJDq**
>
> We thank the reviewer for the valuable feedback and insightful questions! We provide the corresponding clarifications as follows:
>
> ## Q1: Clarification on DGA with EMA (Figure 1,3)
> According to the domain weight evolution trajectories shown in Figure 3(a), DGA without EMA consistently up-weights a specific domain for a certain number of steps before shifting to another. This cyclical behavior not only amplifies overfitting over time but also optimize the model parameters to an ill-positioned location. Also, the first identified relevant domain to be significantly down-weighted when the model shifts to the second domain.
> Notably, the spikes in domain weights align closely with the spikes observed in the validation loss curves, highlighting the correlation between these dynamics and the model's performance fluctuations.
>
> ## W1: Potential explanations for the limited improvement in reasoning accuracy
> We agree with the reviewer that despite a substantial improvement above uniform baseline, DGA only matches or does slightly worse than importance sampling baselines in reasoning accuracy scores, as shown in Table 1.
>
> As for the gap between language modeling ability and reasoning accuracy, we provide two potential explanations in **Appendix B.3.1**: (1) since the target objective in the outer-loop ($L_{spe}$ in Equ. 3) is the next token prediction loss for language modeling, the language modeling performance is expected to be improved. However, it does not necessarily translate to improved reasoning accuracy (2) MMLU is considered to be a challenging reasoning task, where the accuracies can hardly be improved when the model capacity is below some specific threshold (Figure 11. (G) in [1]). Understanding the loss-accuracy gap and deriving an optimization objective that directly benefits the reasoning capability would be a compelling future direction.
>
> Finally, we would appreciate if the reviewer could clarify what they mean by “only an alternative to importance sampling”. Indeed, both DGA and importance sampling pursue the same goal.
>
> ## W2: Confusion in the understanding of "online"
> We call this method “online” because, unlike proxy-based method (e.g. DoGE), it estimates the domain weights on the fly during a single optimization loop. We have clarified this in the manuscript in L57-L59.
>
> ## W3: Assumption of training a LLM with gradient
> We agree with the reviewer that DGA requires the ability to compute gradients of the model. As the reviewer says, we believe that this is not a strong requirement because DGA is used during pre-training or continued pre-training, which are phases typically requiring the ability to compute gradients of the model.
>
> We hope our answers can resolve most of the concerns. If you have any further questions, don’t hesitate to contact us. We will improve the manuscript according to your valuable suggestions!
>
> [1] Emergent Abilities of Large Language Models

---

> ### Comment · Reviewer_eJDq · 2024-11-23
>
> > Finally, we would appreciate if the reviewer could clarify what they mean by “only an alternative to importance sampling”. Indeed, > both DGA and importance sampling pursue the same goal.
>
> As both solve the same problem and DGA performs very comparable to Importance Sampling there's unfortunately little that makes it more attractive except for slightly different application properties. So, it must be viewed as equal alternative to Importance Sampling.

---

> > ### Author Response · Authors · 2024-12-02
> > **End of the discussion period**
> >
> > Dear reviewer eJDq,
> >
> > Thank you for taking the time to review our paper and for engaging in the insightful discussion about embedding-based importance sampling and gradient-based DGA methods. We appreciate your constructive feedback and are pleased to provide detailed comparisons and clarifications below:
> >
> > - (1) **DGA stabilizes training under limited resources**: As demonstrated in Figure 1, importance sampling leads to severe overfitting under constrained training resources, whereas DGA with EMA effectively stabilizes the training process.
> >
> > - (2) **DGA can utilize fine-grained cluster information**: Importance sampling struggles to effectively leverage extremely fine-grained cluster information (Figure 4a) and exhibits significant overfitting with limited samples from the target task (Figure 4b). In contrast, DGA mitigates these issues.
> >
> > - (3) **DGA makes better Generalist and specialist**: Compared to importance sampling, DGA enables the model to achieve a better balance as both a generalist and a specialist (Figure 2).
> >
> > We acknowledge that DGA and importance sampling show comparable performance in terms of accuracy. We believe that using an objective in the outer-loop which directly optimizes the accuracy could be a potential solution. We leave it as a future follow-up of this work.
> >
> > Since tomorrow is the last day of the author-reviewer discussion phase, we hope you can take some time to look into our rebuttals and additional results. Let us know if you have other concerns and questions!
> >
> > Sincerely,
> >
> > the authors

---

> ### Author Response · Authors · 2024-11-23
>
> Thanks for your reply.
>
> Indeed, in some settings, DGA and importance sampling perform quite similarly. However, we also highlight several settings where DGA outperforms importance sampling, for instance, in Figure 1 and Figure 2. So, in our view, there are several scenarios where DGA brings something new compared to importance sampling.

---

### Official Review · Reviewer_TRYF · 2024-11-04

**Soundness:** 2
**Presentation:** 2
**Contribution:** 2
**Rating:** 5
**Confidence:** 3

**Summary:**

This paper introduces Dynamic Gradient Alignment (DGA) as a stable and scalable data mixing method for LLM pretraining to address two key challenges of online domain reweighting. DGA is an online algorithm that adjusts the training data distribution according to the current model status, relying on gradient alignments to estimate progress on the target task. The paper demonstrates that DGA with EMA can significantly mitigate overfitting and yield superior performance on the end-task by balancing exploitation and exploration under limited tokens within generic domains. Additionally, a novel distribution reweighting strategy is proposed, enabling DGA to scale up to extremely fine-grained data domains without incurring intensive computations. Experiments on MMLU show that DGA effectively leverages fine-grained domain knowledge to balance specialty and diversity during training, demonstrating the scalability and efficiency of gradient-alignment-based data reweighting methods in data-constrained settings.

**Strengths:**

1. The paper introduces Dynamic Gradient Alignment (DGA), a novel online gradient alignment algorithm that dynamically adjusts training data mixtures.
2. The experiments are well-designed, covering key scenarios of small pretraining sets and insufficient specialized data, using the MMLU benchmark.
3. The results clearly demonstrate the advantages of DGA over importance sampling and uniform baselines, with detailed analysis of different model scales and domain granularities

**Weaknesses:**

1. The paper provides a brief description of the theoretical foundation of DGA, lacking in-depth mathematical derivation and theoretical analysis. There is no rigorous proof of convergence and stability for DGA, which raises concerns about the robustness and reliability of the proposed method.
2. The paper does not compare DGA with other advanced domain reweighting methods. This omission makes it difficult to assess the relative performance and advantages of DGA compared to existing methods.
3. Benchmarking against other methods is essential to establish the novelty and superiority of the proposed approach. The absence of such comparisons weakens the paper's contribution to the field.
4. The paper lacks a detailed complexity analysis of the experiments, particularly in Section 3.2. A thorough analysis of the computational complexity and resource requirements would provide better insights into the practical implications of implementing DGA.
5. The paper does not provide a comprehensive analysis of the impact of different hyperparameters on the performance of DGA.
Understanding how sensitive DGA is to various hyperparameters is important for its practical implementation and optimization.

**Questions:**

1. Can you provide more mathematical derivation and theoretical analysis of DGA, including proofs of convergence and stability? This would help in understanding the robustness and reliability of the proposed method.
2. How does DGA compare with other advanced domain reweighting methods (e.g., DoGE)? A detailed comparison would help in understanding the relative advantages and limitations of DGA.
3. Can you provide a detailed complexity analysis of the experiments in Section 3.2?
When k is much larger than T_r, 1) is DGA much more complicated than importance sampling method and 2) no better than DoGE in terms of both performance and complexity? Understanding the computational complexity and resource requirements is crucial for evaluating the practical implications of implementing DGA.
4. Can you provide a comprehensive analysis of the impact of different hyperparameters on the performance of DGA? Such ablation study is important for practical implementation and optimization.
5. In Table 1, DGA does not show significant improvement above importance sampling. However, DGA is much more complicated under this experiment setting?
6. Do you have more benchmarking results against other methods? it is essential to establish the novelty and superiority of the proposed approach.
7. In experiments section, there should be a comparison between DGA and DoGE in terms of performance and computational complexity? As DGA is heavily inspired by DoGE, a detailed comparison would provide insights into the relative strengths and weaknesses of both methods.
8. How does DGA perform across different model scales? Can you provide more detailed results and analysis for various model sizes to understand its scalability and effectiveness?
9. What are the potential directions for future work and improvements on DGA? Are there any planned extensions or enhancements to address the current limitations and expand its applicability?

---

> ### Author Response · Authors · 2024-11-22
> **Author response to reviewer TRYF [1/2]**
>
> We thank the reviewer for the thoughtful feedback! We summarize them into 7 questions and provide our clarifications as follows.
>
> ## W1&Q1: Proof of convergence
> As the reviewer points out, the theoretical understanding of data mixing (i.e. domain reweighting) algorithms is a valuable topic that could significantly enhance the theoretical contributions of our paper.  We are happy to report that we could prove that DGA converges in a simple setting. We have updated the paper to include a convergence proof of DGA in a simple convex scenario **(Theorem 2)**. To the best of our knowledge, this is the first theoretical proof of convergence for a data reweighting algorithm. For instance, prior works such as [1,2] do not prove the convergence of their algorithms.
>
> Extending this proof to more realistic scenarios on large language model training would be an intriguing and complex direction for future research. As discussed in the updated version, we believe it would be challenging due to the inherent non-convexity of the problem.
>
> ## W2&W3&Q2&Q6&Q7: Comparison between DGA and DoGE
> We have updated the paper and included the comparison between DGA and DoGE in **Appendix F (Fig. 15)**. The results demonstrate that DGA outperforms DoGE in both specialized loss and general loss. This is because Doge requires training a proxy model before retraining. We believe that this illustrates nicely the merits of our online method.
>
> ## W4&Q3: Detailed complexity comparison between distribution reweighting and domain reweighting
> We agree with the reviewer that the original domain reweighting algorithm cannot handle the case when the number of domains is much larger than the update frequency ($k>>T_r$), as we mentioned in the text L342-362. The distribution reweighting algorithm (Sec. 3.2, Equ. 9) is proposed to resolve this challenge by approximating domain weights $\alpha_{domain} \in R^{k}$ as a linear combination of $N$ distributions, where ($N<<k$). In our experiments, we use $k\in \{64,4096,262k\}, N=23, T_r=100$ which induces $(N+1)/T_r = 24%$ more computation overheads. In comparison, the standard domain reweighting will lead to $(k+1)/T_r = 262k$ overhead factor, which is impossible for LLM training.
>
> We add one more sentence for clarification in **L358-359**.
>
> ## W5&Q4: Ablations on hyperparameters
> We provide the ablations on the EMA factor $\beta$ in **Appendix G.2**, under the same setting in Sec. (3.1). We applied $\beta={0.1,0.9}$, where we find $\beta=0.1$ is the optimal value across all token limits. In comparison, a larger $\beta$ will slow down the learning of domain weights, which hurts training efficiency.
>
> Besides, we are running ablation experiments on the step size ($lr_{\alpha}$) and update frequency ($T_r$) of domain weights $\alpha$ on language modeling task. We will have results by the end of the weekend. For each combination of hyperparameters, we will report the GPU time and the specialized loss on the target domain in in Appendix G.2.
>
> ## Q5: Comparison between DGA and Importance sampling
> We agree with the reviewer that despite a marginal improvement above uniform baseline, DGA shows comparable or even lower reasoning accuracy compared to importance sampling baselines, as shown in Table 1. However, we provided the evidence in Figure 1 and Figure 2 that DGA demonstrates superior stability in language modeling than the importance sampling method. We also included a discussion in **Appendix B.1, Figure 12**, which shows that DGA accelerates the task-adaptive training by a large margin compared to importance sampling.
>
> As for the limited improvement in reasoning accuracy, we included two potential explanations in **Appendix B3.1**: (1) since the target objective in the outer-loop ($L_{spe}$ in Equ. 3) is the next token prediction loss for language modeling, the language modeling performance is expected to be improved. However, it does not necessarily translate to improved reasoning accuracy (2) MMLU is considered to be a challenging reasoning task, where the accuracies can hardly be improved when the model capacity is below some specific threshold (Figure 11. (G) in [3]). How to derive an optimization objective that directly benefits reasoning accuracy would be a compelling direction for future research.

---

> ### Author Response · Authors · 2024-11-22
> **Author response to reviewer TRYF [2/2]**
>
> ## Q8: DGA performance across model scales
> We present language modeling perplexity scores across three model scales (125M, 350M, 750M) in Appendix G.2, Figure 17. While the experiments on the largest models (750M) are still running, our current results on Arxiv demonstrate that DGA consistently outperforms uniform sampling by a significant margin across all three model scales. While it exhibits some degradation in
> general knowledge compared to uniform sampling, the gap narrows on the largest model (750M),
> indicating the potential scalability benefits of DGA in larger model regimes.
>
> ## Q9: Potential future direction of data mixing
> As the reviewer pointed out, the reasoning capability would not necessarily increase if we apply the next token prediction loss as the objective for data mixing optimization. How to derive an optimization objective that directly benefits the reasoning accuracy would be a compelling future direction.
>
> We hope our answers can resolve most of your concerns. If you have any further questions, don’t hesitate to contact us. We will improve the manuscript according to your valuable suggestions!
>
> [1] Doge: Domain reweighting with generalization estimation
> [2] Doremi: Optimizing data mixtures speeds up language model pretraining
> [3] Emergent Abilities of Large Language Models

---

> > ### Author Response · Authors · 2024-11-25
> > **Follow-up response to reviewer TRYF**
> >
> > We have finished the experiments to complete our previous answer.
> >
> > ## W5&Q4: Ablations on hyperparameters
> > We present the ablation results on the step size ($lr_{\alpha}$) and reweight frequency ($T_r$) in **Appendix G.3**, with a detailed discussions on how to determine these hyperparameters.
> >
> > ## Q8: DGA performance across model scales
> > We update the pdf with the complete results on model scaling experiments in **Appendix G.2**.
> >
> > Please let us know if you have any further concerns or questions. We will do our best to address your concerns and improve the manuscript based to your suggestions!

---

> ### Author Response · Authors · 2024-12-02
> **End of the discussion period**
>
> Dear reviewer TRYF,
>
> Thank you again for your efforts reviewing our paper and providing valuable feedback! Since **tomorrow is the last day of the author-reviewer discussion phase**, we hope you can take some time to look into our rebuttals and additional results. Let us know if you have other concerns and questions!
>
> Sincerely,
>
> the authors

---

> > ### Comment · Reviewer_TRYF · 2024-12-03
> >
> > Thanks the authors for their response! I have no further questions.

---

> > > ### Author Response · Authors · 2024-12-03
> > >
> > > Dear Reviewer TRYF,
> > >
> > > Thanks for the response, and we are happy that our additional results have resolve all your concerns! We will appreciate it if you can further consider adjust the scores to the paper. Thanks!
> > >
> > > Sincerely,
> > >
> > > the authors

---

### Official Review · Reviewer_yT5T · 2024-11-05

**Soundness:** 3
**Presentation:** 3
**Contribution:** 2
**Rating:** 5
**Confidence:** 4

**Summary:**

The authors propose a novel pre-training data mixing method, called DGA, which enables LLMs to adapt to downstream tasks with just a few examples. By alternately updating model parameters and domain weights, DGA achieves exploration and exploitation on the training data, thereby solving the overfitting problems of importance sampling. DGA provides an approach for the data mixing problem in pre-training for specific downstream tasks. Additionally, the authors validated their method on the Pile and MMLU datasets, demonstrating DGA’s effectiveness in language modeling for downstream tasks.

**Strengths:**

1,DGA effectively addresses the overfitting issue of importance sampling in scenarios with limited data.

2, By incorporating the EMA term, DGA effectively addresses the issue of training instability.

3, By combining with importance sampling, DGA resolves the issue of "scaling linearly with the number of domains k". Even when there are a large number of domains, it can also demonstrate its advantages.

**Weaknesses:**

1, DGA's reasoning accuracy on datasets like MMLU is not good, but the authors only emphasize that DGA has strong capabilities in language modeling. The performance on downstream tasks should be related to reasoning accuracy, rather than the capability of language modeling.

2, The authors did not compare the performance differences between DGA and other methods under the paradigm of pre-training + fine-tuning or pre-training + ICL, but LLMs typically adapt to downstream tasks using these paradigms, which significantly differs from actual usage scenarios.

**Questions:**

1, How would DGA perform under the paradigms of pre-training + fine-tuning or pre-training + ICL?

---

> ### Author Response · Authors · 2024-11-22
> **Author response to reviewer yT5T**
>
> We thank the reviewer for the feedback and insightful questions! We provide corresponding clarifications as follows:
>
> ## W1: Limited improvement in reasoning accuracy
> We agree with the reviewer that despite a substantial improvement above uniform baseline, DGA performs comparably as importance sampling baselines in reasoning accuracy scores, as shown in Table 1.
>
> As for the considerable gap between language modeling ability and reasoning accuracy, we included an discussion in Appendix B.3.1 with two potential explanations: (1) since the target objective in the outer-loop ($L_{spe}$ in Equ. 3) is the next token prediction loss for language modeling, the language modeling performance is expected to be improved. However, it does not necessarily translate to improved reasoning accuracy (2) MMLU is considered to be a challenging reasoning task, where the accuracies can hardly be improved when the model capacity is below some specific threshold (Figure 11. (G) in [1]). How to derive an optimization objective that directly benefits the reasoning accuracy would be a compelling future direction.
>
> In addition, we report the perplexity scores as the common measurement on language modeling capability. As shown in **Appendix B.1, Figure 12**, DGA achieves up to $6.5\times$ acceleration on task-adaptive training on MMLU benchmark.
>
> ## Q1&W2: Performance comparison on pretraining + finetuning
> We thank the reviewer for the great suggestion! Though the current work mainly focus on the task-adaptive pretraining, finetuning could have a great impact in task adaptation performance in real cases.
> We are currently running the finetuning experiments and we expect to present the results by the end of the weekend.
>
> We hope our answers can resolve most of the concerns. If you have any further questions, don’t hesitate to contact us. We will improve the manuscript according to your valuable suggestions!
>
> [1] Emergent Abilities of Large Language Models

---

> > ### Author Response · Authors · 2024-11-25
> > **Follow-up response to reviewer yT5T**
> >
> > We have finished the experiments to complete our previous answer.
> > ## Q1&W2: Performance comparison on pretraining + finetuning
> > We have included the results after finetuning in **Appendix G.1and Figure 16**, where we show that DGA consistently outperforms the uniform sampling and importance sampling baselines across various scales of fine-tuning tokens from the downstream task. Since the model scale is too small to emerge in-context learning ability, we leave it for future work with scaled-up models.
> >
> > Please let us know if you have any further concerns or questions. We will do our best to address your concerns and improve the manuscript based to your suggestions!

---

> ### Author Response · Authors · 2024-12-02
> **End of the discussion period**
>
> Dear reviewer yT5T,
>
> Thank you again for your efforts reviewing our paper and providing valuable feedback! Since **tomorrow is the last day of the author-reviewer discussion phase**, we hope you can take some time to look into our rebuttals and additional results. Let us know if you have other concerns and questions!
>
> Sincerely,
>
> the authors

---

### Official Review · Reviewer_HaQd · 2024-11-06

**Soundness:** 2
**Presentation:** 3
**Contribution:** 2
**Rating:** 5
**Confidence:** 5

**Summary:**

This paper proposes Dynamic Gradient Alignment (DGA), an online domain reweighting approach for the purpose of identifying an optimal data mixture for training an LLM for a downstream task with access to only a few examples (denoted specialized set).  DGA estimates stepwise optimal domain weights during model training.  The traditional importance sampling approaches suffer from some limitations, such as overfitting into small-scale pretraining data in each domain, difficulty to handle large domain granularity, and easily trapping into quite suboptimal local optimum with limited specialized set. The authors empirically show that DGA alleviates these limitations, offers an effective approach for optimizing training data mixture for downstream tasks with limited specialized set.

**Strengths:**

(1)	The proposed DGA shows advantages over traditional importance sampling approaches in two challenging scenarios, that is, the number of training samples in each domain is limited instead of the unrealistic assumption of unlimited number of tokens, and the domain granularity is quite large making traditional domain reweighting methods cause intractable computational overheads.

(2)	The paper proposed two major innovations in DGA, including incorporating EMA term into domain reweighting to mitigate overfitting and improve stability, and also introducing a distribution reweighting scheme, with the number of distributions smaller than the number of domains, for scaling up to very fine-grained domains (very large number of domains). Though DGA is built upon DoGE, the differentiators from DGA over DoGE are clearly explained in the paper, as the lower variance, smaller overhead, and the novel EMA in DGA, compared to DoGE.

(3) Overall, the paper is clearly written. The Appendix provides useful additional results for the two challenging scenarios, hyperparameter information, and algorithm details.

**Weaknesses:**

(1)	For empirical validations, the paper used Redpajama-v2 as the pre-training data, which has 30T tokens after filtering and deduplication. This is non-trivial amount of data. However, the paper pre-trained models with 125M, 350M, and 750M number of parameters. In comparison, some SOTA LLMs are trained on smaller or comparable amount of data with much larger model sizes, for example, Llama 3 405B trained on 15.6T tokens, QWen2.5-8B trained on 18T tokens, OpenLLaMA 3B, 7B, and 13B trained on 1T tokens, etc. The model sizes explored in this paper may be too small and too limited model capacity to sufficiently learn from 30T tokens data. It would be useful to pre-train models with larger sizes, e.g., to 3B and 7B, which will correspondingly impact the experimental results and analysis.  A related question is, why not use OpenLLaMA setup including its open-source data for the experiments?

(2)	Figure 1 analyzes the effect of DGA and DGA with EMA. However, 30M tokens/domain are still quite large. How about the comparison when the number of tokens/domain is further reduced, e.g., less than 1M, less than 100K, 10K?

(3) One critical question when adapting pre-trained LLM for downstream tasks is how much the resulting model can achieve a good balance between general capabilities and performance on downstream tasks. For example, for instruction following capabilities on a downstream task, how much the model can perform for general instruction following tasks with DGA compared to the baselines. This issue of catastrophic forgetting has not been discussed in the paper.

**Questions:**

(1)	Please address the points listed under Weaknesses.

(2)	There are some typos and grammar errors in the paper. For example, “obtain obtain”.

---

> ### Author Response · Authors · 2024-11-22
> **Author response to reviewer HaQd**
>
> We thank the reviewer for their insightful questions and suggestions which help us improve our work! We provide our clarifications as follows:
>
> ## W1: Model architecture and scales
> We want to clarify that we do not pre-train models on the entire redpajama-v2 (30T tokens), which would induce an intensive computation cost and also be wasteful for these model sizes according to neural scaling law. We train our largest model (N=750M) on ~25B tokens, which is above the standard 20 tokens per parameter derived from scaling laws [1].
> As discussed in the paper, L204-L210 (computational cost and memory overhead), there is in principle no obstacle to scale DGA to larger models given sufficient computation resources. However, pre-training 3B or larger models sadly exceeds our computational budget.
>
> As for the architecture, we adopt a standard decoder-only transformer, which is commonly used in most LLM papers as well as prior works on data mixing [2,3,4]. Since the proposed DGA method is derived from general optimization theory, it should be architecture-agnostic and applicable to OpenLLaMA as well. Seeing how it adapts to other generic pretraining sets with various data quality is an interesting future research direction.
>
> ## W2: More restricted token constraints
> We consider 30M tokens per domain as a strict restriction since this gives 30M*64 = 1.92B tokens in total to train the uniform baseline, which utilizes all tokens across 64 domains. This is a sufficient amount of tokens to train a 125M model without overfitting. **A stricter token constraint, such as 1M tokens per domain, would lead to inevitable overfitting**, even from the uniform baseline, which utilizes all available tokens with the best diversity.
>
> Since we aim to investigate whether DGA could strategically sample from various domains to achieve better performance on the target domain without overfitting, we believe that using 1M tokens per domain is too far from a realistic pre-training scenario.
>
> ## W3: Catastrophic forgetting
> We thank the reviewer for this judicious remark. Indeed, catastrophic forgetting is a serious issue for language models, especially in the task-adaptive training.
>
> In response, we summarized our observations in a detailed discussion in **Appendix H**. We investigate the catastrophic forgetting in various contexts: (1) how does DGA preserve general knowledge comparing to importance sampling and DoGE baselines; (2) how does distribution reweighting algorithm performs comparing to domain reweighting algorithm; (3) how would DGA suffer from catastrophic forgetting when applied on various scale of models. We highlight that DGA effectively mitigates catastrophic forgetting compared to other baselines (Importance sampling and DoGE). Specifically, DGA achieves a well-balanced trade-off: it produces a specialized model with low specialization loss on the target domain while maintaining general knowledge, as evidenced by its low loss on the Pile domains (representing broad, diverse domain knowledge) and the direct generic loss from RedPajama-v2.
>
> We hope our answers resolve most of your concerns. If you have any further questions, don’t hesitate to contact us. Your valuable suggestions have helped us improve the manuscript!
>
> [1] Scaling Laws for Neural Language Models
>
> [2] Doge: Domain reweighting with generalization estimation
>
> [3] Doremi: Optimizing data mixtures speeds up language model pretraining
>
> [4] Specialized language models with cheap inference from limited domain data

---

> ### Author Response · Authors · 2024-12-02
> **End of the discussion period**
>
> Dear reviewer HaQd,
>
> Thank you again for your efforts reviewing our paper and providing valuable feedback! Since **tomorrow is the last day of the author-reviewer discussion phase**, we hope you can take some time to look into our rebuttals and additional results. Let us know if you have other concerns and questions!
>
> Sincerely,
>
> the authors

---

### Author Response · Authors · 2024-11-22
**General Response: Summary of the proposed changes to address the reviewers' concerns.**

We sincerely thank all reviewers for their efforts in reading our paper and providing insightful feedback, which helps us improve our work. We appreciate that the novelty and superiority of DGA as an online data mixing method have been recognized by all the reviewers.

We summarize the main concerns raised by the reviewers with our proposed changes as follows:
1. **Q**: Lack of a theoretical convergence analysis of the proposed method. [TRYF, 9LhJ]

    **A: We have updated the paper to include a convergence proof of DGA in a simple quadratic scenario (Theorem 2).** To the best of our knowledge, this is the first theoretical proof of convergence for a data reweighting algorithm. For instance, prior works such as [1,2] do not prove the convergence of their algorithms.
2. **Q**: Lack of comparison to other domain reweighting baselines (e.g. DoGE). [TRYF, 9LhJ]

    **A: We have updated the paper and include the comparison between DGA and DoGE in Appendix F (Fig. 15).** The results demonstrate that DGA outperforms DoGE in both specialized loss and general loss.
3. **Q**: Lack of discussions on catastrophic forgetting of generic knowledge. [HaQD]

    **A: We provide an in-depth discussion on catastrophic forgetting in Appendix H**, where we show that DGA can preserve more general knowledge during task-adaptive pretraining, while other baselines (Importance sampling, DoGE) are more vulnerable to catastrophic forgetting.
4. **Q**: Lack of ablation studies on hyperparameters. [TRYF]

    **A: We provide the ablations on the EMA factor $\beta$ in Appendix G.2** in domain reweighting with $k=64$, under the same setting in Sec. (3.1). We applied $\beta={0.1,0.9}$, where we find DGA with $\beta=0.1$ can effectively prevent overfitting without sacrificing the learning efficiency on the specilized knowledge. In comparison, a larger $\beta$ will slow down the learning of domain weights, which hurts training efficiency. Meanwhile, **we are running ablation experiments on the step size ($lr_{\alpha}$) and update frequency ($T_r$) of domain weights $\alpha$ on one language modeling task. We will have results by the end of the weekend.** For each combination of hyperparameters, we will report the GPU time and the specialized loss on the target domain in Appendix G.2. However, we found that the hypeparameters discussed in the main text worked well for a broad range of tasks/model sizes.

For other questions and points of clarification, we provide detailed responses correspondingly under each individual review.

---

> ### Author Response · Authors · 2024-11-25
> **Follow-up response:**
>
> We have finished the following experiments to complete our previous response:
>
> ## Q4:  Lack of ablation studies on hyperparameters
> We have included the complete ablation results on two hyperparameters, the step size ($lr_{\alpha}$) and reweight frequency ($T_r$), in **Appendix G.3 and Figure 18**. The results demonstrate that using a step size that is too small will result in slow updates to the domain weights. On the other hand, applying a large step size ($\eta$) can accelerate the learning of domain weights but may also lead to training instability due to overly up-weighted domains. Since EMA can effectively stabilize learning, we recommend that practitioners choose $\eta$ values between 0.1 and 0.5.
> Regarding the reweight frequency, using a smaller $T_r$ generally improves the final specialized loss but increases computation costs. To balance cost and performance, we recommend setting $T_r$ between 30 and 100. However, these values should be tailored to specific use cases and levels of domain granularity.
>
> ## Q5:  Lack of a comparison of model performance after finetuning
> We have included the results after fine-tuning in **Appendix G.1and Figure 16**, where we show that DGA consistently outperforms the uniform sampling and importance sampling baselines across various scales of fine-tuning tokens from the downstream task. In other words, the gain incurred by using DGA during pre-training is kept during fine-tuning.

---

### Meta-Review · Area_Chair_LNS3 · 2024-12-19

**Metareview:**

The paper proposes Dynamic Gradient Alignment (DGA), an online domain re-weighting algorithm designed to optimize training data mixtures for large language models (LLMs) in downstream tasks with limited examples. The method addresses limitations of traditional importance sampling approaches, such as overfitting to small-scale data, difficulty handling fine-grained domains, and suboptimal local minima. The paper claims following contributions: 1) DGA dynamically adjusts domain weights during training by aligning gradients with the target task, enabling effective exploration and exploitation of training data. Exponential moving averages (EMA) is used to stabilize weight updates and improve training scalability. The method is capable of handling extremely fine-grained data domains without significant computational overhead, and adapting to diverse datasets, including mixes of small-scale datasets with high diversity. 2) ablations and empirical results on benchmarks such as MMLU and the Pile showed the efficacy of proposed method; overfitting is mitigated with balances specialty and diversity during training.


Strength of this paper
- Tackles challenges in recently popular topics in LLMs area such as overfitting and instability when limited training samples are available. The problem has strong utility for real-world scenarios with limited resources.
- Empirically validate the efficacy of proposed method. Using benchmarks like MMLU to validate DGA’s effectiveness compared to importance sampling and uniform baselines, across varying model scales, domain granularities, and data limitations.
- The proposal of using online gradient alignment of dynamically adjusting training data mixtures , and exponential moving average to stablize is reasonable


Weakness of this paper

Several reviewers raised few concerns/limitations of this paper. By addressing these limitations, the paper could strengthen its experiment and expand impact.

- Experiment scope is limited. The paper only compare with standard importance sampling, but not advanced domain reweighting methods or other variations. The model sizes (125M, 350M, 750M parameters) are too small to fully leverage the 30T tokens in the Redpajama-v2 dataset. Larger models, such as 3B or 7B, should be explored to evaluate the method's effectiveness, and verify whether the effectiveness can be translated to models with sizes commonly used in the community. Results also lack exploration of smaller token/domain scenarios (e.g., <1M, <100K, or 10K tokens/domain), which would better illustrate DGA's adaptability under extreme data constraints. The paper also need to evaluate how DGA impacts instruction-following capabilities or balances general and task-specific performance without forgetting general capabilities. Comparisons with common paradigms such as pre-training + fine-tuning or pre-training + in-context learning (ICL) should also be conducted.
- Some concerns about generalizability. E.g., the paper shows DGA improves language modeling but does not consistently enhance downstream reasoning capabilities (e.g., MMLU); the paper does not provide sensitivity analysis of hyperparameters, which is essential for optimization in real-world scenarios. There is limited mathematical derivation or discussion about training convergence to validate stability and robustness of proposed method; the assumptions behind DGA, such as reliance on gradient-based LLM training, limit its applicability compared to more general sampling techniques.

**Additional Comments On Reviewer Discussion:**

In addition to above weaknesses, reviewers also raised some other weaknesses (e.g., clarity in manuscript and terminology, stability of performance without EMA, and computational complexity) and improvements during rebuttal. Although some of the weakness have been improved / somewhat addressed during rebuttal session (e.g., further explanation on the concerns raised by reviewers, more experiment results added), overall review rating was not raised significantly, and the rating is still at borderline.  I think the session is too short and some weaknesses are hard to address in such a short period of time (e.g., comprehensive and scale of experiments). I would like to see a more comprehensive modification to systematically working on these suggestions. Given the high bar of ICLR, I think the paper is still of limited interests to the audience , and thus I recommend to reject the paper, and the authors to re-work on these weakness and re-submitting to future conferences.

---

### Decision · Program_Chairs · 2025-01-22

Reject